# Mechanistic Insights into Mancozeb-Induced Redox Imbalance and Structural Remodelling Affecting the Function of Human Red Blood Cells

**DOI:** 10.3390/antiox14111274

**Published:** 2025-10-23

**Authors:** Sara Spinelli, Elisabetta Straface, Lucrezia Gambardella, Giuseppina Bozzuto, Daniele Caruso, Angela Marino, Silvia Dossena, Rossana Morabito, Alessia Remigante

**Affiliations:** 1Department of Chemical, Biological, Pharmaceutical and Environmental Sciences, University of Messina, 98122 Messina, Italy; saspinelli@unime.it (S.S.); marinoa@unime.it (A.M.); rmorabito@unime.it (R.M.); 2Biomarkers Unit, Center for Gender-Specific Medicine, Istituto Superiore di Sanità, 00161 Rome, Italy; elisabetta.straface@iss.it (E.S.);; 3National Center for Drug Research and Evaluation, Istituto Superiore di Sanità, 00161 Rome, Italy; 4Complex Operational Unit of Clinical Pathology, Papardo Hospital, 98158 Messina, Italy; 5Institute of Pharmacology and Toxicology, Research and Innovation Center Regenerative Medicine & Novel Therapies, Paracelsus Medical University, 5020 Salzburg, Austria; silvia.dossena@pmu.ac.at; 6Department of Biomedical, Dental and Morphological and Functional Imaging, University of Messina, 98122 Messina, Italy

**Keywords:** xenobiotic, oxidative stress, anion exchanger 1, estrogen receptors, erythrocy

## Abstract

Mancozeb is a broad-spectrum fungicide used extensively in agriculture to protect crops against a wide range of plant diseases. Although its capacity to induce oxidative stress is well documented, the cytotoxic effects of mancozeb on red blood cells (RBCs) remain poorly characterized. The present study aimed to investigate the cytotoxic effects of mancozeb on isolated RBCs, with particular focus on oxidative stress-induced cellular and molecular alterations. Human RBCs were exposed to mancozeb (0.5–100 µM) for 24 h. No hemolytic activity was observed across the tested concentrations. However, 10 and 100 µM mancozeb induced a significant increase in intracellular reactive oxygen species (ROS), leading to lipid and protein oxidation and impaired Na^+^/K^+^-ATPase and anion exchanger 1 (AE1) function. These changes resulted in altered RBC morphology, reduced deformability, and increased methemoglobin levels. Alterations in glycophorin A distribution, anion exchanger 1 (AE1) clustering and phosphorylation, and α/β-spectrin and band 4.1 re-arrangement indicated disrupted membrane–cytoskeleton interactions. A release of extracellular vesicles (EVs) positive for glycophorin A and annexin-V was also observed, consistent with plasma membrane remodeling. Despite increased intracellular calcium, eryptosis remained minimal, possibly due to activation of protective estrogen receptor (ER)-mediated pathways involving ERK1/2 and AKT signaling. Activation of the cellular antioxidant system and the glutathione redox system (GSH/GSSG) occurred, with catalase (CAT) playing a predominant role, while superoxide dismutase (SOD) activity remained largely unchanged. These findings offer mechanistic insights regarding the potential health impact of oxidative stress induced by pesticide exposure.

## 1. Introduction

Mancozeb [[1,2-ethanediylbis] carbamodithioate]] (2-)] is a mixture of manganese with [[1,2-ethanediylbis [carbamodithioate]] (2-) zinc belonging to the group of dithiocarbamate (DTC) fungicides and more specifically to the class of compounds known as ethylene bisdithiocarbamates (EBDCs) used to protect potato and tomato crops. In 2020, the European Commission issued a regulation refusing to renew the substance approval, largely based on concerns about its classification as toxic to reproduction. However, while the European Union forbids the use of these pesticides for products cultivated in Europe, it allows imported products from third countries to contain residues of such substances. In these countries (the United States, Brazil, India, China, and Japan), mancozeb is frequently used to protect crops from fungal diseases that could compromise the quality standards required for export. Without the use of such a pesticide, the quality of agricultural products could be negatively affected, potentially impacting trade relationships and reducing revenues [1]. As informed by the World Health Organization (WHO), approximately 70 thousand workers die from acute and chronic poisoning by pesticides, and another 7 million develop non-fatal disease [2]. In such a context, industrial and agricultural workers accumulate mancozeb through inhalation or skin contact and represent the main high-risk population for mancozeb poisoning [3,4,5]. In addition, according to World Bank data, every year 355,000 people die of involuntary poisoning by pesticides [6,7]. This figure includes individuals who come into contact with pesticide-contaminated food [8], particularly in developing countries where pesticide regulations and food safety practices are less stringent. Thus, food contamination remains a significant route of exposure to mancozeb [9,10,11]. Over time, mancozeb is rapidly metabolized in the environment and within organisms into ethylenethiourea (ETU) in the presence of water and oxygen. Although mancozeb is insoluble in water and thus unlikely to contaminate groundwater directly, its metabolite ETU is water-soluble and has the potential to leach through the soil, increasing the risk of groundwater contamination [12,13]. Thus, ETU is the main metabolite of mancozeb that is commonly detected in the environment [14]. Although DTCs are characterized by short persistence in the environment, causing mild acute toxicity upon exposure, mancozeb and ETU are known to have additional long-term toxic effects of primary concern, due to their isothiocyanate skeleton; such molecular moiety may interfere with enzymes containing sulfhydryl groups, thereby affecting biological systems [15,16]. This has been linked to an increased risk of reproductive disorders, endocrine disruption, cancer, and respiratory problems [5,17,18,19].

In humans, different reports have shown that mancozeb can reach the blood circulation, inducing an increase in oxidative stress levels or alterations in the reactive oxygen scavenging system [3,20]. In the presence of toxic chemicals and environmental pollutants, red blood cells (RBCs) are particularly prone to oxidative damage due to the presence of high levels of polyunsaturated fatty acids in their membranes and high cellular concentrations of hemoglobin, an iron-containing metalloprotein that can bind to oxygen molecules [21,22,23,24]. The structural and functional simplicity of these cells makes them a convenient cellular model that is especially suitable for studies of the toxicity of xenobiotics [21,22,25,26]. It is well known that human RBCs are responsible for carrying oxygen throughout the body; thus, xenobiotic-induced morphological or biochemical alterations may prove to be lethal [27]. In particular, the RBC plasma membrane is made of a phospholipid bilayer, integral membrane proteins, and cytoskeleton-associated proteins; this structure is elastic and confers suitable shape and flexibility to the RBCs [22,28,29,30]. Different xenobiotics, including mancozeb, can alter RBC rheological properties by increasing reactive species, which can lead to damaged lipids and proteins and accelerate membrane micro-vesiculation, which limits the ability of RBCs to maintain their shape [15,20,31]. An excess of intracellular reactive oxygen species (ROS) can provoke a cascade of events, including hemoglobin oxidation and heme degradation, depletion of the antioxidant defense system, and increased intracellular calcium levels, resulting in anion exchanger 1 (AE1, *SLC4A1*) tyrosine phosphorylation, culminating in an increased susceptibility of RBC to premature removal from blood circulation [32]. Therefore, it is very important to obtain mechanistic insights into the consequences of exposure of human RBCs to xenobiotics.

Here, a detailed study has been performed on the cellular and molecular mechanisms associated with RBC exposure to mancozeb, with a focus on AE1 function, which is closely related to maintaining cellular and systemic homeostasis. Obtained data show that mancozeb induces oxidative stress-related derangements in the absence of haemolysis by impairing both phosphorylation pathways and the intracellular antioxidant defense system, resulting in damage to cellular components, which in turn adversely affects RBC structure and function. Importantly, mancozeb was studied in its original form because its metabolite ETU does not fully reflect the compound chemistry, and in in vitro use of ETU may not reproduce the primary toxic effects on RBCs, including membrane damage and oxidative stress. Moreover, the concentrations of mancozeb used in this study are not directly comparable to those detected in the blood of healthy human donors [33], as the levels observed in individuals with either direct or indirect exposure are significantly lower [2,3]. This discrepancy is expected, considering that mancozeb undergoes systemic metabolism, leading to the formation of various byproducts, such as ETU [3,33]. As the present study was conducted using a cell-based model, we selected a higher range of concentrations to ensure measurable responses. This experimental approach is consistent with previous findings reported by Quds and colleagues [20].

## 2. Materials and Methods

### 2.1. Solutions and Chemicals for Human RBC Sample Processing

All chemicals were purchased from Sigma (Milan, Italy). Stock solutions of mancozeb (CAS number: 8018-01-7, Sigma-Aldrich, Italy, purity 99.7%, 1 mM) and 4,4′-diisothiocyanatostilbene-2,2′-disulfonate (DIDS, 10 mM) were prepared in dimethyl sulfoxide (DMSO); NaNO_3_ (20 mM) and 50 mM 3-amino-1,2,4-triazole (3-AT; 3 M) were dissolved in distilled water. The H_2_O_2_ experimental solution was obtained by diluting a 30% *v*/*v* stock solution in distilled water, whereas 2,2′-Azobis (2-methylpropionamidine) dihydrochloride (AAPH, 0.5 M) was dissolved in PBS (pH 7.4). Both ethanol and DMSO never exceeded 0.001% *v*/*v* in the experimental solutions and were previously tested on human RBCs to exclude possible hemolytic damage.

### 2.2. Preparation of Human RBC Samples

The research protocol received approval from the Institutional Ethics Committee of the University of Messina, Italy (protocol 52–22). All procedures were performed with donor consent and in compliance with the principles of the Declaration of Helsinki. Venous blood from volunteer donors from Caucasian ethnicity (aged 20–55 years) was collected into ethylenediaminetetraacetic acid (EDTA) tubes. Human RBCs were first rinsed in an isotonic buffer composed of 150 mM NaCl, 5 mM HEPES, and 5 mM glucose, adjusted to pH 7.4 with an osmolarity of 300 mOsm/kgH_2_O [34]. Cells were then centrifuged (Neya 16R, 1200× *g* for 5 min) to remove plasma and the buffy coat. Subsequently, the erythrocytes were resuspended in the same isotonic solution, at a hematocrit of 3%, according to the specific experimental protocols described below.

#### 2.2.1. Haemolysis Measurement

To verify the % haemolysis, RBCs were incubated with or without mancozeb (0.5–100 µM) in isotonic solution, then suspended at 0.5% haematocrit in isotonic solution, centrifuged (Neya 16R, 1200× *g*, 5 min), and resuspended at 0.05% haematocrit in a 0.9% *v*/*v* NaCl solution [35,36]. Haemoglobin free (supernatant) absorbance was measured at 405 nm wavelength and subtracted for the absorbance of the blank (0.9% *v*/*v* NaCl solution).

#### 2.2.2. Detection of Apoptotic RBCs

Red blood cells were left untreated or exposed to mancozeb (0.5–100 µM)-containing isotonic solution and processed to detect apoptosis by using the FITC-conjugated Annexin V apoptosis detection kit (Biovision, Milpitas, CA, USA) and Trypan blue staining (0.05% Trypan blue for 10 min at room temperature) [37]. Then, erythrocytes were analyzed with a FACScan flow cytometer (Becton-Dickinson, Mountain View, CA, USA) equipped with a 488 nm argon laser.

### 2.3. Analysis of Cell Shape by Scanning Electron Microscopy (SEM)

Cell morphology was assessed in human RBCs incubated with 10 µM or 100 µM mancozeb for 24 h at 37 °C. After treatment, cells were harvested, placed on poly-L-lysine–coated slides, and fixed for 20 min at room temperature using 2.5% glutaraldehyde in 0.1 M cacodylate buffer (pH 7.4). Post-fixation was carried out with 1% OsO_4_ in 0.1 M sodium cacodylate buffer. Samples were then dehydrated through a graded ethanol series (30% to 100%). Absolute ethanol was progressively replaced with a 1:1 mixture of hexamethyldisilazane (HMDS) and ethanol, followed by pure HMDS. After complete removal of HMDS, the specimens were dried in a desiccator. The dried preparations were mounted on stubs, sputter-coated with a 10 nm layer of gold, and examined using a Cambridge 360 scanning electron microscope (Leica Microsystems, Wetzlar, Germany). Morphological alterations were quantified by evaluating at least 500 erythrocytes per condition (50 cells per SEM field at 3000× magnification), using three independent samples.

### 2.4. Detection of Extracellular Vesicle Shedding

To isolate extracellular vesicles (EVs), RBCs were left untreated or treated with 10 µM or 100 µM mancozeb for 24 h, at 37 °C. Following treatment, samples were centrifuged at 900× *g* for 5 min at 25 °C, in order to separate the cell pellet from the supernatant. For each sample, the supernatant was further centrifuged at 13,000× *g* for 10 min to sediment the potential EVs. The pellet containing EVs was incubated with a mouse monoclonal anti-glycophorin A antibody (1:500; Sigma-Aldrich, G7900) for 30 min at room temperature. Successively, after three washes with PBS (pH 7.4), the pellet was incubated with anti-mouse IgG (H+L) Highly Cross-Adsorbed Secondary Antibody (1:500; Alexa Fluor™ 488, Waltham, MA, USA). Finally, after three washes in PBS (pH 7.4), the EVs were processed with the eBiosciences^TM^ APC-conjugated Annexin-V Kit (Thermo Fisher Scientific, Waltham, MA, USA) and analyzed by flow cytometry (FACS). Fluorescence intensity values provide a semi-quantitative analysis. In addition, EV volume was measured by FACS using forward scatter (FSC) parameters. The percent change (Δ%) in volume was calculated using the following formula: Δ% = (V_treated_ − V_untreated_/V_untreated_) ×100, where V_treated_ represents the mean FSC value of treated samples and V_untreated_ represents the mean FSC value of control samples.

### 2.5. Evaluation of Intracellular Calcium Content

Intracellular calcium levels were evaluated in human RBCs left untreated or treated with 10 µM or 100 µM mancozeb for 3 and 24 h, at 37 °C. Following treatment, cells were incubated with 4 µg/mL Fluo-3 AM (Molecular Probes), a calcium-sensitive fluorescent dye, for 45 min at 37 °C to allow intracellular dye loading. After incubation, RBCs were washed twice with PBS (pH 7.4) to remove excess dye. Intracellular calcium was then measured using a FACSCalibur flow cytometer (BD Biosciences, Heidelberg, Germany) equipped with a 488 nm argon laser for excitation. For each sample, a minimum of 20,000 events were collected to ensure statistical reliability. Data is presented as fluorescence arbitrary units, reflecting intracellular calcium levels.

### 2.6. Detection of Deformability Measured by Elongation Index

Erythrocyte deformability was assessed by ektacytometry (LORRCA; Mechatronics Instruments BV, AN Zwaag, The Netherlands) in human RBCs incubated for 24 h at 37 °C with either 10 µM or 100 µM mancozeb, as well as in untreated controls. The procedure followed the method reported by Donadello et al. [38]. The elongation index (EI) was calculated using the formula EI = (L − W)/(L + W), where *L* and *W* represent the major and minor axes of the diffraction pattern, respectively. Increased EI values reflect greater cellular deformability at any applied shear stress. EI curves were generated across 12 distinct shear stress levels. Since deformability reaches a plateau at 50 Pa, the maximum elongation index (EImax) was derived from the corresponding portion of the curve.

### 2.7. Measurement of Na^+^/K^+^-ATPase Activity

Na^+^/K^+^-ATPase activity was evaluated in untreated RBCs and in cells exposed to 10 µM or 100 µM mancozeb for 24 h at 37 °C by quantifying the amount of inorganic phosphate (Pi) released from ATP, as previously described [39,40]. Erythrocytes were adjusted to a hematocrit of 3% (1 mL), lysed in 15 mM Tris-HCl (pH 7.4), and centrifuged at 13,000× *g* for 15 min at 4 °C to eliminate hemoglobin. The membrane pellets were then resuspended in 250 μL of distilled water, and total protein content was measured using the Bradford assay [41]. For the enzymatic reaction, two aliquots of 100 μL from each sample were incubated for 1 h at 37 °C with 900 μL of either reaction buffer A (50 mM Tris-HCl, 4 mM MgCl_2_, 3 mM ATP-Na_2_, pH 7.4) or reaction buffer B (120 mM NaCl, 50 mM Tris-HCl, 20 mM KCl, 4 mM MgCl_2_, 3 mM ATP-Na_2_, pH 7.4). The reaction was stopped by adding 200 μL of 50% (*v*/*v*) trichloroacetic acid. Color development was achieved by incubation with 2% (*w*/*v*) ascorbic acid for 20 min at room temperature. Absorbance was then recorded at 725 nm using a UV-21 Onda spectrophotometer (Carpi (MO), Italy). Enzyme activity was calculated by subtracting the absorbance of buffer A from that of buffer B and expressed as Pi per mg of protein.

### 2.8. Assessment of Oxidative Stress Parameters

#### 2.8.1. Detection of ROS Levels

The ROS levels were evaluated by the cell-permeable indicator 2′,7′-dichlorofluorescein diacetate (H_2_DCFDA, D6883, Sigma-Aldrich) in RBCs left untreated or treated with 10 µM or 100 µM mancozeb for 24 h at 37 °C, according to the manufacturer’s instructions. As positive control, human RBCs were incubated with 20 mM H_2_O_2_ at 25 °C for 30 min. Reactive oxygen species (ROS) production was quantified using a microplate reader (Fluostar Omega, BMG Labtech, Ortenberg, Germany) set at 485 nm for excitation and 535 nm for emission. Background fluorescence was subtracted from all measurements, as reported in [42]. Results are expressed in arbitrary units.

#### 2.8.2. Measurement of TBARS Levels

Levels of thiobarbituric acid (TBA)-reactive substances (TBARS) were measured as reported by Mendanha and colleagues [43]. Human RBCs, left untreated or treated with 10 µM or 100 µM mancozeb for 24 h at 37 °C, were suspended in isotonic solution (3% hematocrit, 100 µL), treated with 200 µL SDS 8.1%, 1.5 mL of 20% (*v*/*v*) acetic acid, and 1 mL TBA 1%, and the mixture was incubated at 95 °C for 30 min. As positive control, an aliquot of RBCs was incubated with 50 mM AAPH for 1 h at 37 °C. Sample absorbance was measured at 532 nm (Onda spectrophotometer, UV-21). Results are indicated as µM TBARS levels (1.56 × 10^5^ M^−1^ cm^−1^ molar extinction coefficient).

#### 2.8.3. Measurement of Total Sulfhydryl (-SH) Groups

The quantification of -SH groups was carried out according to the method described by Aksenov and Markesbery [44]. Human RBCs, either untreated or exposed to 10 µM or 100 µM mancozeb for 4 h at 37 °C, were centrifuged (Neya 16R, 1200× *g*, 5 min), and 8.5 µL of the resulting pellet was lysed in 1 mL of distilled water. From this lysate, 20 μL was mixed with 940 μL of phosphate-buffered saline (PBS, 0.1 M, pH 7.4) containing 1 mM EDTA. The reaction was initiated by adding 30 μL of 50 mM 5,5′-dithiobis (2-nitrobenzoic acid) (DTNB), and samples were incubated for 40 min at 25 °C in the dark. Parallel control samples lacking either DTNB or cell lysate were processed under identical conditions. Following incubation, absorbance was recorded at 412 nm using an Onda UV-21 spectrophotometer. The concentration of 3-thio-2-nitrobenzoic acid (TNB) was calculated after subtracting the background absorbance of DTNB-only blanks. For the positive control, an aliquot of RBCs was treated with 50 mM AAPH for 1 h at 37 °C to fully oxidize thiol groups. Results were expressed as μM TNB per mg of protein.

#### 2.8.4. Measurement of Methemoglobin (MetHb) Content

The methemoglobin (MetHb) levels were determined as reported by Naoum and colleagues [45]. The method relies on spectrophotometric detection of MetHb and oxyhemoglobin at 630 nm and 540 nm, respectively. Human RBCs (3% hematocrit), either untreated or exposed to 10 µM or 100 µM mancozeb for 24 h at 37 °C, were used. A 100 μL aliquot of the cell suspension was lysed in 6 mL of hypotonic buffer (15 mM NaH_2_PO_4_, 10 mM KH_2_PO_4_) supplemented with 100 μL of 1% SDS (hemolysate A). Then, 300 μL of hemolysate A was further diluted in 3 mL of the same hypotonic solution (hemolysate B). To generate a fully oxidized reference sample, a separate aliquot of RBCs was incubated with 4 mM NaNO_3_ for 1 h at 25 °C, a known inducer of MetHb formation. Absorbance readings of hemolysates A and B were taken at 630 nm and 540 nm, respectively, using an Onda UV-21 spectrophotometer. Methemoglobin levels were calculated using the following formula: % MetHb = (OD 630 nm × 100)/(OD 630 nm + (OD 540 × 10)).

#### 2.8.5. Measurement of Intracellular Free Iron

Intracellular free iron levels were measured in human RBCs incubated for 24 h at 37 °C with either 10 µM or 100 µM mancozeb, as well as in untreated controls. The analysis was performed using a commercial colorimetric kit (Kamiya Biomedical Company, cat. no. KT-757). The method is based on the reduction of dissociated iron, which then forms a colored complex with the ferrozine chromogen. The intensity of the resulting chelate, detected at 560 nm, is directly proportional to the iron content in the sample [46].

### 2.9. Measurement of Cytosolic Syk Kinase Activity

Phosphorylated SYK (phospho-Tyr SYK) levels in RBCs were measured using a commercially available Phospho-SYK (Tyr) ELISA kit (RAB0999, Sigma-Aldrich) following the manufacturer’s instructions. Briefly, human RBCs were isolated from whole blood by centrifugation at 1000× *g* for 10 min at 4 °C, followed by removal of the plasma and buffy coat. Red blood cells (3% hematocrit) were then lysed in the lysis buffer provided with the kit (supplemented with phosphatase and protease inhibitors) and incubated on ice for 30 min. Lysates were centrifuged at 13,000× *g* for 15 min at 4 °C, and the supernatants were collected. Equal amounts of total protein (typically 50–100 µg per well) were loaded on ELISA plate wells pre-coated with capture antibodies specific for phospho-SYK. After incubation at room temperature for 12 h, wells were washed, and HRP-conjugated detection antibody was added. Following incubation and additional washes, TMB substrate solution was added, and the reaction was stopped with 1 N sulfuric acid. Absorbance was measured at 450 nm with a reference at 570 nm using a microplate reader (EZRead 400 ELISA, BioChrom, Cambridge, UK). Data were expressed as absorbance.

### 2.10. Preparation of Erythrocyte Plasma Membrane Proteins

RBCs (3% hematocrit), either untreated or exposed to 10 or 100 µM mancozeb for 24 h at 37 °C, were resuspended in ice-cold hypotonic Tris-HCl buffer (20 mM, pH 7.5) supplemented with a protease inhibitor cocktail. Samples were then centrifuged at 13,000× *g* for 15 min at 4 °C (Neya 16R). Plasma membrane proteins were isolated following the procedure described by Pantaleo et al. [47]. Briefly, cells were repeatedly washed and centrifuged under the same conditions in hypotonic buffer containing protease inhibitors to remove hemoglobin, which remained in the supernatant. The resulting membrane fraction was treated with 1% (*v*/*v*) SDS and kept on ice for 20 min to achieve solubilization. The obtained membrane protein extracts were stored at −80 °C. Protein concentrations were determined using the Bradford assay [41].

#### SDS-PAGE Preparation and Western Blotting Analysis

Plasma membrane proteins were denatured for 10 min at 95 °C after solubilization in Laemmli buffer [48]. The proteins were then separated by SDS-polyacrylamide gel electrophoresis and transferred onto a polyvinylidene fluoride (PVDF) membrane under constant voltage for 2 h. Membranes were blocked with BSA for 1 h at 25 °C and incubated overnight at 4 °C with primary antibodies diluted in TBST: mouse monoclonal anti-AE1 (B9277) or anti-pTyR (T1325), both from Sigma-Aldrich, at dilutions of 1:5000 and 1:1000, respectively. Afterwards, membranes were incubated for 1 h at 25 °C with peroxidase-conjugated secondary antibodies—rabbit anti-mouse or goat anti-rabbit IgG (A9044 and A0545, Sigma-Aldrich)—diluted 1:10,000 in TBST. To ensure equal protein loading, the same membrane was probed with mouse monoclonal anti-β-actin (A1978, Sigma-Aldrich, 1:10,000), as described by Yeung et al. [49]. Chemiluminescent detection was performed using the SuperSignal West Pico substrate (Pierce Thermo Scientific, Rockford, IL, USA), and protein bands were quantified by densitometry using Bio-Rad (Hercules, CA, USA) ChemiDoc™ XRS+ and analyzed with Image Quant TL v2003.

### 2.11. Analytical Cytology

Human RBCs, left untreated or treated with 10 or 100 µM mancozeb for 24 h at 37 °C, were fixed in 3.7% formaldehyde in PBS (pH 7.4) for 10 min at room temperature. After washing with the same buffer, cells were permeabilized with 0.5% Triton X-100 (Sigma-Aldrich) in PBS (pH 7.4) for 5 min at room temperature. Following a PBS wash, the samples were incubated for 30 min at 37 °C with the following mouse monoclonal antibodies: anti-AE1 (1:500; B9277, Sigma-Aldrich), anti-α/β-spectrin (1:500; sc-271130, Santa Cruz Biotechnology, Dallas, TX, USA), anti-glycophorin A (1:500; G7900, Sigma-Aldrich); anti-ankyrin (1:500; Invitrogen, Waltham, MA, USA, 33–8800); anti-4.1 (1:500; Santa Cruz Biotechnology, sc 398983), anti-ERα (1:500; Santa Cruz Biotechnology, sc 8002); anti-Erβ (1:500; Santa Cruz Biotechnology, sc 53494); anti-phosphorylated ERK1/2 (1:500; BD Transduction Laboratories, Franklin Lakes, NJ, USA, 612358); and anti-phosphorylated AKT (1:500; Santa Cruz Biotechnology, sc 271966). Successively, all samples were washed thrice in PBS (pH 7.4) and incubated for 30 min at 37 °C with anti-mouse IgG (H+L) Highly Cross-Adsorbed Secondary Antibody (1:500; Alexa Fluor™ 488). All samples were then washed three times with PBS (pH 7.4) and incubated for 30 min at 37 °C with an anti-mouse IgG (H+L) Highly Cross-Adsorbed Secondary Antibody conjugated to Alexa Fluor™ 488 (1:500). Fluorescence analysis was performed using an Olympus BX51 Microphot microscope or a FACScan flow cytometer (Becton Dickinson, Mountain View, CA, USA) equipped with a 488–544 nm argon laser. A minimum of 20,000 events per sample were acquired. Fluorescence intensity histograms are presented to provide a semi-quantitative assessment of staining analysis.

### 2.12. Measurement of SO_4_^2−^ Uptake

Anion exchanger 1 activity was determined as the uptake of SO_4_^2−^ in human RBCs left untreated or treated with 10 µM or 100 µM mancozeb for 24 h at 37 °C, as previously reported [21,50,51,52,53,54,55]. Briefly, after washing, human RBCs were resuspended in 35 mL of SO_4_^2−^ medium (composition in mM: Na_2_SO_4_ 150, HEPES 5, glucose 5; pH 7.4; osmolarity 300 mOsm/kg H_2_O) and incubated at 25 °C for 5, 10, 15, 30, 45, 60, 90, and 120 min. At each time point, 10 µL of DIDS (10 µM), an AE1 inhibitor [56], was added to 5 mL aliquots, which were immediately kept on ice; then samples were washed three times with cold isotonic solution and centrifuged (Neya 16R, 4 °C, 1200× *g*, 5 min) to remove extracellular SO_4_^2−^. The cell pellet was lysed by adding distilled water, and proteins were precipitated using 4% (*v*/*v*) perchloric acid. After centrifugation (Neya 16R, 4 °C, 2500× *g*, 10 min), the supernatant containing the SO_4_^2−^ accumulated by RBCs during the incubation period was analyzed turbidimetrically. For the assay, 500 µL of the supernatant was mixed sequentially with 500 µL of glycerol diluted 1:1 in distilled water, 1 mL of 4 M NaCl, and 500 µL of 1.24 M BaCl_2_·2H_2_O. Absorbance was measured at 425 nm (Onda Spectrophotometer, UV-21). A calibration curve, prepared by precipitating known SO_4_^2−^ concentrations in a separate experiment, was used to convert absorbance values to [SO_4_^2−^] in L cells × 10^−2^. The SO_4_^2−^ uptake rate constant (min^−1^) was calculated using the equation: C_t_ = C_∞_ (1 − e^−rt^) + C_0_, where C_t_, C_∞_, and C_0_ indicate the intracellular SO_4_^2−^ concentrations measured at time t, ∞, and 0, respectively, e represents the Neper number (2.7182818), r indicates the rate constant of the process, and t is the specific time at which the SO_4_^2−^ concentration was measured. The rate constant corresponds to the inverse of the time required to reach approximately 63% of the total intracellular SO_4_^2−^ [50]. Data are reported as [SO_4_^2−^] L cells × 10^−2^, representing the micromolar concentration of SO_4_^2−^ internalized by 10 mL RBCs at 3% hematocrit.

### 2.13. Endogenous Antioxidant Activity Assessment

#### 2.13.1. Catalase Activity Assay

Catalase (CAT) activity was evaluated by the catalase assay kit (MAK381, Sigma-Aldrich, Milan, Italy) in cells left untreated or treated with 10 µM or 100 µM mancozeb for 24 h at 37 °C. As positive control, cells were treated with 50 mM AAPH for 1 h at 37 °C. Catalase activity was determined by reading the absorbance at 570 nm wavelength (Fluostar Omega, BMG Labtech, Ortenberg, Germany) after subtracting the background absorbance.

#### 2.13.2. Superoxide Dismutase Activity Assay

Superoxide dismutase (SOD) activity was evaluated by the SOD activity assay kit (CS0009, Sigma-Aldrich, Milan, Italy) in cells left untreated or treated with 10 µM or 100 µM mancozeb (24 h at 37 °C) or 50 mM 3-AT, a specific inhibitor of CAT [57], for 15 min at 25 °C. As positive control, cells were treated with 50 mM AAPH for 1 h at 37 °C. SOD activity was determined by reading the absorbance at 450 nm wavelength (Fluostar Omega, BMG Labtech, Ortenberg, Germany) after subtracting the background absorbance.

#### 2.13.3. GSH/GSSG Ratio Measurement

GSH/GSSG ratio was quantified by the GSH assay kit (MAK440, Sigma-Aldrich, Milan, Italy) using an enzymatic recycling method with glutathione reductase in cells that were untreated or treated with 10 µM or 100 µM mancozeb for 4 h at 37 °C. As positive control, cells were treated with 50 mM AAPH for 1 h at 37 °C. Sample absorbance was measured at 412 nm (Fluostar Omega, BMG Labtech, Ortenberg, Germany). Results are expressed as a GSH/GSSG ratio.

### 2.14. Experimental Data and Statistics

All data are presented as the arithmetic mean ± standard error of the mean (S.E.M.). Statistical analyses and graphical representations were performed using GraphPad Prism (version 8.0, GraphPad Software, San Diego, CA, USA) and Microsoft Excel (Version 2019, Redmond, WA, USA). Differences between group means were assessed using one-way or two-way analysis of variance (ANOVA) followed by Bonferroni’s post hoc test, or by Student’s *t*-test where appropriate. A *p*-value of less than 0.05 was considered statistically significant. The symbol (*n*) indicates the number of biological replicates.

## 3. Results

### 3.1. Detection of Cellular Morphology in Mancozeb-Treated RBCs

As reported in Figure 1, exposure of RBCs to both 10 µM and 100 µM mancozeb for 24 h induced marked morphological changes. In particular, SEM analysis revealed a significant presence of leptocytes, namely RBCs exhibiting a flattened morphology, accounting for 41% and 48% of the cells, respectively (Table 1). Notably, treatment with 100 µM mancozeb also resulted in a marked reduction in RBC volume, further indicating compromised cellular shape (Table 1).

### 3.2. Detection of Vesicle Shedding in Mancozeb-Treated RBCs

After 24 h of incubation with mancozeb, flow cytometric analysis revealed a concentration-dependent increase in extracellular vesicle sub-populations. In particular, flow cytometric profiling identified specific EV sub-populations, including vesicles positive for glycophorin A, annexin-V, and a double-positive (^+^) subset (Figure 2A–D). Specifically, EVs positive for glycophorin A accounted for 10% and 15% of total EVs in samples treated with 10 µM and 100 µM mancozeb, respectively (Figure 2A). The percentage of annexin V-positive vesicles was 30% and 65% in the same treatment conditions (Figure 2B), while vesicles positive for both glycophorin A and annexin V reached 9.5% and 13.7%, respectively (Figure 2C). In addition, such vesiculation was also accompanied by a concomitant decrease in vesicle volume, as determined by forward scatter (FSC) measurements (Figure 2E–G; Table 2). Notably, the mean volume of EVs released from mancozeb-treated cells was significantly lower than that of EVs derived from untreated cells. To further quantify this effect, the percentage change (Δ%) in EV volume relative to the control was also calculated (Table 2).

#### 3.2.1. Detection of Glycophorin A Expression Levels and Distribution in Mancozeb-Treated RBCs

Flow cytometry analysis revealed a significant reduction in glycophorin A expression levels in human RBCs incubated with 10 µM or 100 µM mancozeb for 24 h at 37 °C compared to untreated controls (Figure 3A). Immunofluorescence analysis detected changes in glycophorin A distribution upon the plasma membrane (Figure 3B). Specifically, in cells treated with both concentrations of mancozeb, glycophorin A appeared redistributed and clustered along the plasma membrane (indicated by arrows) with respect to untreated RBCs (Figure 3B).

#### 3.2.2. Detection of Intracellular Calcium Levels in Mancozeb-Treated RBCs

Results revealed a significant increase in intracellular calcium levels in RBCs treated with 10 or 100 µM mancozeb after 24 h of incubation (Figure 4). On the contrary, no significant increase in intracellular calcium was observed after 3 h incubation (Figure 4).

### 3.3. Detection of Cellular Deformability in Mancozeb-Treated RBCs

As shown in Figure 5, human RBCs exposed to both 10 µM and 100 µM mancozeb for 24 h at 37 °C exhibited significantly reduced deformability. At each shear stress value tested, the elongation index (EI) of treated cells was markedly lower than that of untreated cells. Specifically, RBCs incubated with 10 µM mancozeb showed a maximum EI (EI max) of 0.50, compared to 0.65 in untreated cells. Similarly, cells treated with 100 µM mancozeb exhibited an EI max of 0.45, versus 0.67 in the untreated samples. No statistically significant difference between samples treated with 10 µM or 100 µM mancozeb was reported.

#### Assessment of Na^+^/K^+^ ATPase Activity in Mancozeb-Treated RBCs

In RBCs exposed to both 10 µM and 100 µM mancozeb, the activity of the Na^+^/K^+^ ATPase pump was significantly lower than that detected in untreated cells (Figure 6). Na^+^/K^+^-ATPase pump activity did not show a significant difference between cells treated with 10 µM and 100 µM mancozeb.

### 3.4. Assessment of Oxidative Stress Parameters in Mancozeb-Treated RBCs

#### 3.4.1. Evaluation of TBARS Levels

Figure 7A shows TBARS levels in cells treated with 10 µM or 100 µM mancozeb for 24 h at 37 °C, as well as in cells exposed to 50 mM AAPH for 1 h at 37 °C. As expected, treatment with the pro-oxidant AAPH, used as a positive control, resulted in a significant increase in TBARS levels compared to untreated cells. Similarly, TBARS levels were significantly higher in mancozeb-treated samples compared to controls. Notably, this increase was dose-dependent, with higher TBARS levels observed in cells treated with 100 µM mancozeb compared to those treated with 10 µM.

#### 3.4.2. Evaluation of Total-SH Group Content

Figure 7B shows the total content of -SH groups in human RBCs treated with 10 or 100 µM mancozeb for 24 h at 37 °C or, alternatively, in cells exposed to 50 mM AAPH for 1 h at 37 °C. As expected, incubation with AAPH, used as positive control, led to a significant reduction in the content of the -SH groups compared to untreated cells. Sulfhydryl group content detected in samples exposed to 10 µM or 100 µM mancozeb was significantly lower than that measured in untreated cells. Notably, this reduction was dose-dependent, with a greater decrease in -SH groups observed in cells treated with 100 µM mancozeb compared to those treated with 10 µM.

#### 3.4.3. Detection of MetHb Levels and Intracellular Iron Release in Mancozeb-Treated RBCs

Figure 8A reports MetHb levels detected in human RBCs exposed to 10 µM or 100 µM mancozeb for 24 h at 37 °C. The MetHb levels measured after exposure of RBCs to a well-known MetHb-forming compound (4 mM NaNO_3_ for 1 h at 25 °C) were significantly higher than those detected in untreated cells. In parallel, MetHb levels measured in samples incubated with 100 μM mancozeb were also significantly higher than those measured in untreated samples. Instead, the treatment with 10 μM mancozeb did not induce a significant increase in MetHb production. Figure 8B reports the amount of intracellular free iron. In samples treated with 100 µM mancozeb, intracellular free iron levels were higher than those observed in both 10 µM mancozeb-treated and control RBCs.

### 3.5. Measurement of Cytosolic Syk Kinase Activity in Mancozeb-Treated RBCs

Figure 9 reports the activity of cytosolic Syk kinase in human RBCs exposed to 10 µM or 100 µM mancozeb for 24 h at 37 °C. A significant decrease in Syk kinase activity was observed in cells treated with 100 µM mancozeb compared to untreated controls. No difference in samples exposed to 10 µM mancozeb with respect to untreated cells has been reported.

#### Detection of Phosphorylation and AE1 Expression Levels in Mancozeb-Treated RBCs by Western Blot

Figure 10A shows the AE1 expression levels detected in cells treated with 10 µM or 100 µM mancozeb for 24 h at 37 °C. In both conditions, a significant reduction in AE1 protein expression levels has been reported. In contrast, only exposure to 100 µM mancozeb provoked an increase in tyrosine phosphorylation of AE1 protein (Figure 10B).

### 3.6. Detection of AE1 Expression Levels and Distribution in Mancozeb-Treated RBCs by Flow Cytometry

Flow cytometry showed that the levels of AE1 significantly decreased in human RBCs incubated with 10 or 100 µM mancozeb for 24 h with respect to untreated cells (Figure 11A). Moreover, significant differences were detected between samples treated with 10 or 100 µM mancozeb, respectively. Anion exchanger distribution was also assessed by immunofluorescence. In particular, AE1 was mainly clustered (arrows) in samples incubated with 10 µM or 100 µM mancozeb for 24 h with respect to untreated RBCs (Figure 11B).

#### Anion Exchanger-Mediated SO_4_^2−^ Uptake in Mancozeb-Treated RBCs

To evaluate the anion exchange capability of AE1, the SO_4_^2−^ uptake was determined in cells exposed to 10 µM or 100 µM mancozeb for 24 h at 37 °C (Figure 12). In control cells, SO_4_^2−^ uptake progressively increased, reaching equilibrium in 17.71 min (the rate constant of SO_4_^2−^ uptake is 0.056 ± 0.005 min^−1^; Table 1). In cells treated with 10 µM mancozeb, the transport rate constant (0.079 ± 0.004 min^−1^; Table 1) was higher than that detected in control cells, thus denoting accelerated transport kinetics. On the contrary, the transport rate constant in cells treated with 100 µM mancozeb (0.043 ± 0.005 min^−1^; Table 3) was significantly decreased with respect to untreated cells. In both experimental conditions, a significant decrease in SO_4_^2−^ amount internalized by RBCs after 45 min of incubation in SO_4_^2−^ medium was reported (Table 3). As expected, in DIDS-treated cells, the rate constant of SO_4_^2−^ uptake and the SO_4_^2−^ amount internalized were substantially reduced compared to control cells (Table 3).

### 3.7. Determination of Cytoskeleton-Associated Proteins in Mancozeb-Treated RBCs

The distribution of cytoskeleton-associated proteins was evaluated by flow cytometry and immunofluorescence analysis in cells incubated with 10 µM or 100 µM mancozeb for 24 h at 37 °C. In RBCs exposed to both mancozeb doses, the fluorescence signal of ankyrin, α/β-spectrin, as well as band 4.1 was not significantly different than that of samples left untreated (Figure 13A–C). However, an intense rearrangement and redistribution with the formation of peripheral clusters (red arrows) of specific cytoskeleton-associated proteins, especially α/β-spectrin and band 4.1, in mancozeb-treated samples was detected (Figure 13D). On the contrary, no marked change in the ankyrin redistribution on the plasma membrane has been observed.

### 3.8. Measurement of ERα/β Content and Distribution in Mancozeb-Treated RBCs

The levels of ERα/β, reported as mean values of fluorescence intensity obtained by flow cytometry and detected in human RBCs treated with 10 µM or 100 µM mancozeb for 24 h at 37 °C were comparable to those reported in untreated samples (Figure 14A). Figure 14B,C report representative immunofluorescence images showing changes in ERα/β distribution in the presence of both mancozeb doses at the level of the RBC plasma membrane. In particular, both ER clusterization (arrows) at the level of the plasma membrane in cells exposed to both mancozeb concentrations was observed compared to untreated cells (Figure 14B,C). This clustering suggests a potential translocation of ERα/β from the cytosol to the plasma membrane in response to mancozeb exposure.

### 3.9. Measurement of Phosphorylated ERK and AKT Content in Mancozeb Treated-RBCs

Figure 15A,B show the levels of phosphorylated ERK1/2 (pERK1/2) and AKT (pAKT) detected by flow cytometry in human RBCs left untreated or treated with 10 µM and 100 µM mancozeb for 24 h. The exposure of cells to mancozeb induced a significant increase in phosphorylation of both kinase proteins compared to untreated samples. Notably, pERK1/2 expression levels appeared to increase in a dose-dependent manner, with higher levels observed in cells treated with 100 µM mancozeb compared to what was observed with 10 µM mancozeb. In parallel, cells treated with both mancozeb concentrations also exhibited significantly higher pAKT levels compared to those measured in untreated cells.

### 3.10. Evaluation of the Endogenous Antioxidant Capacity in Mancozeb-Treated RBCs

SOD and CAT activity, as well as the GSH/GSSG ratio, were measured as an estimate of the endogenous antioxidant system of the human RBCs exposed to 10 µM or 100 µM mancozeb for 24 h at 37 °C (Figure 16). As expected, the treatment of human RBCs with 50 mM AAPH (1 h, at 37 °C) significantly increased both SOD and CAT activity. Parallelly, exposure to 50 mM AAPH (1 h at 37 °C) led to a reduction in the GSH/GSSG ratio, thus reflecting an increase in GSSG levels and/or a decrease in GSH levels. In RBC samples incubated with both doses of mancozeb, the activity of SOD was not significantly different with respect to that of untreated samples. However, the SOD activity increased in RBC pre-treated with 50 mM 3-AT, a specific and irreversible inhibitor of CAT, and then exposed to 10 μM or 100 μM mancozeb (Figure 16A). In addition, an increased CAT activity and decreased GSH/GSSG ratio in mancozeb-treated RBCs were reported (Figure 16B,C).

## 4. Discussion

Given their simple structure and lack of organelles, human RBCs represent a reliable model for studying the toxicological impact of various xenobiotics, including the fungicide mancozeb. However, the specific toxic effects of mancozeb and its metabolites on RBCs remain poorly characterized. In the present study, we then investigated their potential impact on isolated human RBCs, focusing on oxidative stress-induced damage and functional alterations. The preliminary phase of the study was designed to assay a wide range of mancozeb concentrations (0.5–100 µM), in order to exclude any overt cytotoxic effects on RBC integrity, using a specific incubation period (24 h). None of the tested concentrations induced detectable hemolysis (Appendix A), and were therefore deemed appropriate for further investigation. Subsequently, we evaluated whether these non-hemolytic concentrations could induce increased oxidative stress. To this end, intracellular reactive oxygen species (ROS) levels were quantified in RBCs following 24 h of mancozeb exposure (Appendix A). A significant increase in ROS production was observed within the concentration range of 10 to 100 µM mancozeb. Based on this observation, the lowest and the highest concentrations associated with an oxidative response (10 µM and 100 µM, respectively) were selected for subsequent analyses.

Different xenobiotic compounds, including mancozeb, coming in contact with human RBCs may have preferential molecular targets leading to oxidative damage and eventually affecting the function of the whole cell [15,20,25]. Relevant structural modifications in the cellular shape were detected by SEM after exposure of RBCs to mancozeb. A significant number of RBCs lost their canonical biconcave shape and displayed a flattened morphology (Figure 1 and Table 1). Any changes in RBC shape can lead to a decreased plasma membrane asymmetry and, consequently, to altered cellular integrity [58]. In line with these observations, human RBCs with altered morphology showed a reduction in cell volume (Table 1). The increased fraction of ROS detected in the cells treated with mancozeb (Appendix A) may have reacted with lipids and proteins, increasing their oxidation (Figure 7A,B), which in turn may have affected the plasma membrane structure in terms of fluidity and integrity. Accordingly, reduced cellular deformability, represented by a reduced elongation index, has been found in mancozeb-incubated RBCs (Figure 5). Another important feature related to RBC shape involves Na^+^/K^+^-ATPase, which plays a pivotal role in maintaining cell morphology. In particular, this ion pump plays a crucial role in preserving optimal intracellular cation concentrations, thereby regulating RBC volume, water balance, and the surface area-to-volume ratio, all of which are essential determinants of blood rheology [59]. Consequently, any alterations in the activity of Na^+^/K^+^-ATPase may result in impaired RBC deformability in blood circulation [60]. Structurally, the Na^+^/K^+^-ATPase pump consists of two subunits: (1) the α-subunit (isoforms 1, 2, and 3), which exhibits ATPase activity and contains binding sites for Na^+^ and K^+^; and (2) the β-subunit, which is essential for the correct insertion and stabilization of the complex within the plasma membrane [61,62]. To further elucidate the role of the Na^+^/K^+^-ATPase pump in maintaining RBC deformability under mancozeb-induced oxidative stress, its activity was evaluated (Figure 6). The obtained data revealed a significant decrease in the Na^+^/K^+^-ATPase activity in the mancozeb-treated cells (Figure 6). The reduction in Na^+^/K^+^-ATPase activity could be attributed to a disrupted interaction with plasma membrane components, possibly as a result of enhanced lipid peroxidation (Figure 7A). Consequently, increased lipid peroxidation may impair the proper membrane localization of the Na^+^/K^+^-ATPase, disrupting its structural and functional integrity, thus contributing to reduced membrane fluidity [46,63]. Alternatively, elevated levels of intracellular reactive species may also have promoted increased phosphorylation of the Na^+^/K^+^-ATPase, potentially inducing conformational changes that impair the functional activity [59,64]. However, the decrease in pump activity is not consistently reported in the literature, as the effect may depend on the specific molecular targets primarily affected by oxidative stress. It is also plausible that elevated intracellular reactive species modify the α-subunit through oxidation of sulfhydryl (-SH) groups or protein carbonylation, which could, in some cases, even result in increased Na^+^/K^+^-ATPase activity [61,65]. In sum, increased ROS production elicited by exposure to mancozeb may have altered RBC morphology and deformability at least in part via functional alterations of the plasma membrane and/or ion transporters, consequent to lipid peroxidation and protein oxidation. In summary, the structural and oxidative alterations observed in mancozeb-treated RBCs can collectively compromise RBC function. These modifications may impair membrane integrity, fluidity, and deformability, ultimately affecting the cell ability to maintain optimal volume, ionic balance, and proper circulation in the bloodstream.

In human RBCs, many radical species are generated primarily due to the oxidation of hemoglobin and the release of free iron [66]. Under physiological conditions, human red blood cells contain approximately 3% methemoglobin, as NADH-dependent cytochrome b5 reductase efficiently reduces Fe^3+^ methemoglobin back to Fe^2+^ hemoglobin [67]. However, under conditions of excessive oxidative stress, the generation of ROS surpasses the buffering capacity of the cell’s antioxidant defenses [68]. Thus, in cells exposed to 100 µM mancozeb, the elevated ROS levels likely promoted hemoglobin oxidation to methemoglobin (Figure 8A) and increased intracellular iron (Figure 8B), thereby enhancing RBC vulnerability to premature oxidative damage and impairing their normal physiological functions. Mancozeb most likely impaired RBC metHb reductase activity, leading to an accumulation of metHb that was not efficiently converted back to oxygenated haemoglobin [69,70]. Additionally, the formation of non-functional hemoglobin may impair AE1-mediated oxygen transport, reducing oxygen delivery to tissues and cells throughout the body [71,72,73]. This protein consists of two domains that are distinct both in structure and function. The N-terminal cytoplasmic domain of AE1 contains binding sites for cytoskeletal and cytoplasmic proteins, including hemoglobin [74,75]. In this regard, our results showed that hemoglobin oxidation (Figure 8A) led to a reorganization of AE1 distribution into multiple membrane clusters, possibly due to dimer or oligomer formation (Figure 11B), and caused a significant reduction in overall AE1 expression levels (Figure 10A). Based on these observations, we hypothesize that hemoglobin oxidation may induce structural changes in AE1, promoting its clustering and altering the stability of the plasma membrane. Such perturbations may impair both the biophysical properties of the membrane and the RBC capacity to mediate effective oxygen exchange [23,76].

Anion exchanger 1 (AE1 or SLC4A1) also plays a crucial role in ion transport [77]. It enables the exchange of chloride and bicarbonate (Cl^−^/HCO_3_^−^) across the membrane, thereby facilitating efficient carbon dioxide removal from tissues. This exchange is critical for maintaining systemic acid-base homeostasis and indirectly contributes to effective oxygen delivery throughout the body. The C-terminal domain of AE1 promotes the Cl^−^/HCO_3_^−^ exchange across the plasma membrane [77], and its function can be evaluated through the measurement of the rate constant for sulfate (SO_4_^2−^) uptake [21,52,53,78,79]. This methodological approach is advantageous because the SO_4_^2−^ uptake is slower and more easily measurable than the physiological uptake of Cl^−^ or HCO_3_^−^ [22,51,80]. In RBCs treated with 10 µM mancozeb, the rate constant for SO_4_^2−^ uptake was significantly accelerated compared to untreated controls (Figure 12, Table 3), no change in AE1 phosphorylation was detected (Figure 10B), nor was any hemoglobin oxidation observed (Figure 8A). However, the observed reduction in AE1 proteins (Figure 10A) may have triggered a compensatory response aimed at maintaining RBC functionality under oxidative stress conditions. This adaptive mechanism could involve increased activity of the remaining AE1 proteins (Figure 12, Table 3), likely supported by the simultaneous activation of the cell’s endogenous antioxidant defense system (Figure 16). On the contrary, in RBCs incubated with 100 µM mancozeb, the rate constant for SO_4_^2−^ uptake was significantly reduced compared to control cells (Figure 12, Table 3). The reduction in the transport rate is most likely linked to the formation of oxidized hemoglobin (Figure 8A) and/or AE1 phosphorylation (Figure 10B) following an increase in intracellular ROS induced by mancozeb treatment (Appendix A). In particular, the association between molecules of AE1 and metHb can initiate a cascade of biochemical and/or structural alterations, including the release of microparticles laden with hemichromes [47,81,82] and clustered AE1 regions. However, no hemichrome production was detected by fluorescence microscopy in the present study, suggesting that they are likely sequestered within the vesicles [83]. Thus, these phenomena can not only lead to the loss of plasma membrane components but also to alterations of RBC shape, consistent with the appearance of leptocytes (Figure 1, Table 1). Summarizing, it is tempting to speculate that the dual effect on the anion exchange rate depends on the specific structure targeted by the stressors and the underlying molecular pathways.

Oxidative stress is one of the proposed mechanisms contributing to membrane shedding (vesiculation) [84]. Extracellular vesicles (EVs) are enclosed by a lipid bilayer that differs in composition from the plasma membrane from which they originate. This compositional difference suggests that their formation is governed by a regulated mechanism that selectively enriches or excludes specific molecular components [85]. The EV release fulfills a dual function: (1) facilitating the removal of damaged cellular material and (2) acting as both indicators and mediators of oxidative stress, with relevant implications in pathologies linked to redox imbalance and vascular dysfunction. Severe disruptions of protein–protein interactions, which are associated with altered RBC morphology, promote enhanced vesicle release, though not always accompanied by increased phosphatidylserine exposure. The obtained data have revealed an increased number of EVs in RBCs incubated with both mancozeb concentrations (Figure 2). Specifically, flow cytometric profiling identified vesicles positive for glycophorin A, annexin-V, and a double-positive subset (Figure 2A–C). Recently, two principal mechanisms have been suggested as the main driving forces of membrane shedding and EV production in human RBCs: (1) plasma membrane/cytoskeleton uncoupling, and (2) intracellular calcium accumulation [84,85]. The RBC membrane consists of a lipid bilayer embedded with transmembrane proteins and decorated with glycans on its external (plasma-facing) surface. Beneath the inner leaflet of the bilayer lies the cytoskeleton, a complex protein network that endows the cell with remarkable deformability and mechanical resilience [86,87]. The cytoskeleton is tethered to the lipid bilayer via AE1 at the spectrin–ankyrin binding sites and via glycophorin A at the actin junctional complexes [88,89]. Based on the obtained data, we propose that one of the mechanisms driving vesiculation involves mancozeb-induced oxidative stress, which may enhance the phosphorylation of membrane proteins, particularly AE1, thereby weakening the linkage between the lipid bilayer and the underlying cytoskeleton. This hypothesis is indirectly supported by the observed reduction in AE1 expression levels at the plasma membrane (Figure 10A), along with a concurrent increase in AE1 phosphorylation (Figure 10B) in (100 µM) mancozeb-treated RBCs. In addition, previous studies have confirmed the presence of AE1, glycophorin A, and actin, but not spectrin, within EVs [90], which is partially corroborated by our findings. Specifically, we observed increased levels of glycophorin A in EVs derived from mancozeb-treated RBCs (Figure 2A), while spectrin expression levels remained unchanged following mancozeb exposure in intact RBCs (Figure 13D). Taken together, decreased levels of glycophorin A in RBCs, hemoglobin damage, and altered phosphorylation of membrane proteins such as AE1 may weaken the anchorage between the lipid bilayer and the cytoskeletal network. In particular, disruption of the AE1–ankyrin–spectrin complex is predicted to cause relaxation of the cytoskeletal spring, leading to spontaneous buckling of the lipid bilayer and, consequently, membrane evagination and vesiculation [91]. Moreover, RBCs treated with mancozeb exhibited a decrease in cell volume, consistent with membrane shedding (Table 2). Such a mechanism, frequently observed in RBCs subjected to excessive oxidative stress, acts as a protective response by facilitating the removal of damaged cellular components, thereby contributing to the preservation of cell viability [84].

As mentioned above, a second mechanism that induces vesiculation is represented by the increase in intracellular calcium concentration, which can be triggered by various cytotoxic stimuli, including exposure to xenobiotics [92]. The increase in intracellular calcium triggers biochemical changes, resulting in the activation of floppase/scramblase and inhibition flippase; all enzymes that control the externalization of phosphatidylserine [93]. In this regard, we demonstrated an increase in annexin-V positive vesicle percentage after exposure of RBCs to both mancozeb concentrations (Figure 2B). In addition, the increase in intracellular calcium levels, especially after 24 h incubation with mancozeb (Figure 4), may activate proteolytic enzymes such as calpain. This activation can lead to the cleavage of AE1, a key structural component of the RBC membrane, thereby causing membrane destabilization, cytoskeletal disassembly, and the release of microvesicles. An additional calcium-dependent vesiculation mechanism involves the activation of two membrane proteins, CD59 and stomatin [84,94,95]. CD59 protects RBCs from complement-mediated lysis by inhibiting the formation of the membrane attack complex (MAC) [96]. Instead, stomatin, a membrane-associated protein, contributes to maintaining membrane stability and organization [96]. Both proteins have been observed to cluster within the RBC plasma membrane, a phenomenon believed to result from changes in lipid composition and membrane curvature driven by factors such as altered phospholipid organization and enzymatic activity, including that of sphingomyelinase [97]. Extracellular vesicles generated by sphingomyelinase exhibit both phosphatidylserine exposure and glycophorin A [84,97]. In particular, vesicles positive for glycophorin A and annexin-V accounted for 9.5% and 13.7% of the total population in cells treated with mancozeb, respectively (Figure 2C). Plasma membrane shedding might be either the cause or the outcome of size reduction in human RBCs incubated with mancozeb (Table 1). On one hand, the reduction in size leads to a lower surface-to-volume ratio, which limits gas exchange between RBCs and tissue cells, and increases osmotic fragility, both factors that make the cells more prone to hemolysis [84,98]. On the other hand, however, the formation of such EVs may serve as a rescue mechanism for RBCs by removing altered molecular components, which are well-established markers of early damaged RBCs, thereby prolonging the lifespan of these cells [84]. Vesicles collected during mancozeb treatment were also accompanied by a concomitant decrease in vesicle volume compared to EVs derived from untreated cells (Figure 2E–G; Table 2). Although this phenomenon cannot be mechanistically explained by the data gathered in the present study, this observation points to a difference in protein composition and/or formation mechanism between EVs produced in physiological conditions and following exposure to mancozeb.

These pesticides can be classified as xenoestrogens or endocrine-disrupting chemicals, meaning they are capable of mimicking endogenous estrogens and altering physiological processes [99]. Their estrogenic or anti-estrogenic effects are thought to result from interactions with plasma membrane ERs, as described by Gea and colleagues [100]. In this context, abnormal localization and clustering of estrogen receptors (primarily ERα) at the plasma membrane were observed in cells exposed to mancozeb (Figure 14B,C), without any detectable changes in total ER expression levels (Figure 14A). Plasma membrane-bound ERs play a role in RBC homeostasis by activating multiple intracellular signaling pathways associated with protein kinases such as MAPK/ERK1/2 and PI3K/AKT. These kinases facilitate estrogen-modulated non-genomic signaling [101,102] thereby regulating several physiological processes, including membrane deformability, adhesion to the endothelium, and cell survival. To further investigate the potential effects of mancozeb binding to ERs, we analyzed the phosphorylation status of ERK and AKT kinases. Supporting the notion that ERK1/2 and AKT mediate cellular responses to various stressors [103] including xenoestrogens, our results show that RBC exposure to mancozeb led to increased levels of phosphorylated ERK1/2 and AKT (Figure 15A,B). Specifically, the engagement of membrane-associated ERs triggers kinase pathways that promote RBC survival [104]. These protective responses may partially mitigate the deleterious effects induced by mancozeb exposure. In particular, AKT could phosphorylate and inactivate pro-eryptotic proteins such as glycogen synthase kinase 3 beta (GSK-3β), thereby preventing the activation of eryptotic-like pathways and promoting RBC survival. Furthermore, estrogen signaling could preserve membrane asymmetry by maintaining flippase activity, which prevents phosphatidylserine exposure and thus avoids recognition and clearance of RBCs by macrophages [105,106]. Although our data confirm an increase in intracellular calcium levels in cells exposed to mancozeb (Figure 4), this effect may be modulated by ER activation (Figure 14B). Specifically, estrogen signaling may attenuate calcium influx by modulating calcium-permeable cation channels, such as Gardos channels, thereby preventing the activation of calcium-dependent enzymes like caspase-3, which could irreversibly damage the membrane and cytoskeleton proteins [107]. Consistent with the potential activation of protective mechanisms, our data show no significant loss of cytoskeleton-associated proteins in RBCs incubated with mancozeb, but rather a marked redistribution within the cytoskeletal network (Figure 13D). Together, both pathways contribute to limiting caspase activation. Regarding the MAPK/ERK pathway, its activation via ERs in RBCs orchestrates a protective network through the canonical Ras → Raf → MEK → ERK1/2 kinase cascade [108]. ERK1/2 phosphorylates cytoskeleton-associated proteins, including spectrin, ankyrin, and protein 4.1, strengthening their interactions to maintain membrane integrity and prevent phosphatidylserine exposure and membrane scrambling, which are key steps in eryptosis [109]. This pathway supports flippase activity, preserving membrane asymmetry by actively transporting phosphatidylserine from the outer to the inner leaflet of the plasma membrane. In fact, despite the high proportion of RBCs displaying morphological changes following mancozeb treatment, the percentage of cells undergoing eryptosis remained very low (<0.5%, Appendix A), suggesting that membrane remodelling occurs prior to the activation of the complete eryptotic program [109]. However, it is also important to note that hyperactivation of intracellular mechanisms modulated by ER signaling could lead to phosphorylation and activation of endothelial nitric oxide synthase (eNOS), potentially influencing RBC function through nitric oxide (NO) production. Straface and colleagues have shown that increased nitric oxide (NO) production via the eNOS isoform is dependent on ERK1/2 phosphorylation triggered by excessive stimulation of estrogen receptors [25,104]. Elevated NO levels can react with reactive species to form peroxynitrite anion (ONOO^−^), which can modify protein thiol groups [110]. In human RBCs, peroxynitrite has been reported to cause: (a) alterations in cell shape, (b) abnormal AE1 distribution, (c) reduced expression of glycophorin A at the plasma membrane, (d) major cytoskeletal rearrangements, and (e) formation of methemoglobin. Therefore, ERs may act as a double-edged sword, providing protective effects under certain conditions while potentially triggering detrimental responses when excessively activated.

A significant increase in tyrosine phosphorylation of RBC membrane proteins, mainly represented by AE1 (Figure 10B), was measured following exposure to mancozeb. Among the various protein kinases present in human RBCs, the most well-known is p72Syk (Syk), which is responsible for the phosphorylation of AE1 [111]. However, the cytosolic activity of Syk kinase was significantly reduced by 100 µM mancozeb compared to control conditions (Figure 9). Notably, Syk may translocate from the cytosol to the membrane, facilitating its interaction with membrane-bound AE1 [47]. In this regard, a phosphorylation-driven oxidation mechanism was likely involved, in which Syk binds to oxidized AE1, phosphorylates it, and disrupts membrane–cytoskeleton interactions. Also, this process may have promoted the formation of AE1 clusters (Figure 11B), which may eventually be released within extracellular vesicles. Additionally, mancozeb-induced excessive oxidative stress probably inhibited the expression of protein tyrosine phosphatases (PTPs), resulting in increased tyrosine phosphorylation (Figure 10B) of specific membrane proteins (i.e., AE1), which could reduce the stability of the RBC structure.

Human RBCs also play a key role in maintaining redox balance [66]. Their antioxidant defense system is highly efficient, relying on both enzymatic and non-enzymatic mechanisms. In this study, we evaluated the activity of the main endogenous antioxidant enzymes, SOD) and CAT, as well as the glutathione redox balance (GSH/GSSG ratio) following exposure to mancozeb. Increased ROS levels lead to enhanced CAT activity and a reduction in the GSH/GSSG ratio (Figure 16B and C). Conversely, exposure to mancozeb did not significantly affect SOD activity (Figure 16A). This response suggests that the primary reactive oxygen species generated is hydrogen peroxide (H_2_O_2_), rather than the superoxide anion (•O_2_^−^), which is typically detoxified by SOD via the reaction: 2•O_2_^−^ + 2H^+^ → H_2_O_2_ + O_2_. In contrast, H_2_O_2_ is eliminated through the action of CAT, which catalyzes the reaction: 2H_2_O_2_ → 2H_2_O + O_2_, and by the glutathione system, in which glutathione peroxidase (GPx) reduces H_2_O_2_ using GSH as follows: 2GSH + H_2_O_2_ → GSSG + 2H_2_O [112]. Notably, exposure to 3-amino-1,2,4-triazole (3-AT), a specific inhibitor of CAT [57], resulted in a significant increase in SOD activity in samples treated with mancozeb (Figure 16A). This observation suggests that CAT inhibition leads to the accumulation of H_2_O_2_, which may exceed the detoxification capacity of the GSH/GPx system [113]. As a consequence, the excess H_2_O_2_ can interact with free iron (Figure 8B), initiating Fenton and Haber-Weiss reactions, which result in the formation of highly reactive hydroxyl radicals (•OH) and additional superoxide anions, and contribute to metHb production (Figure 8A) and lead to SOD activation as a homeostatic reaction (Figure 16A). These findings highlight the central role of CAT in mitigating oxidative stress induced by exposure to xenobiotics such as pesticides. In this context, CAT activity serves as a crucial RBC defense mechanism, significantly contributing to the maintenance of intracellular redox homeostasis.

## 5. Conclusions

This study provides compelling evidence that mancozeb, a widely used fungicide, exerts cytotoxic effects on human RBCs, mainly by triggering oxidative stress. These changes compromised membrane integrity, reduced cellular deformability, altered cell morphology, and disrupted ion exchange processes and membrane-cytoskeleton anchorage. Mancozeb-induced oxidative stress also triggered the release of EVs enriched in specific markers (e.g., glycophorin A, annexin-V), suggesting active vesiculation as a compensatory mechanism to preserve RBC viability. The activation of ER-mediated pathways and downstream kinases, such as ERK1/2 and AKT, appeared to provide partial cytoprotective effects. These non-genomic signaling cascades may contribute to the maintenance of membrane integrity and asymmetry, limiting eryptotic responses despite morphological alterations. CAT played a central role in counteracting mancozeb-induced oxidative stress, emphasizing the importance of this endogenous antioxidant enzyme in maintaining redox homeostasis in xenobiotic stress. In conclusion, the present study demonstrates that even in the absence of hemolysis, mancozeb induces profound sub-lethal alterations in RBC structure and function through oxidative mechanisms. These findings further underscore the relevance of RBCs as a sensitive and informative model for evaluating the cytotoxic potential of environmental pollutants and illuminate the molecular mechanisms underlying health risks linked to pesticide exposure.

## Figures and Tables

**Figure 1 antioxidants-14-01274-f001:**
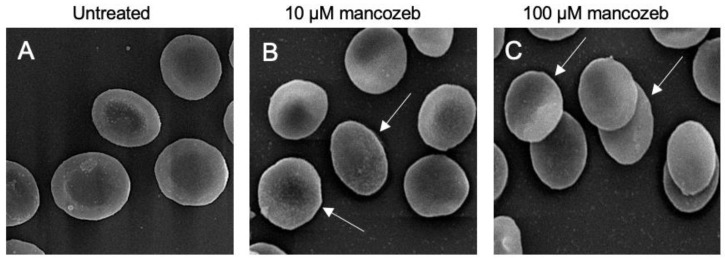
Evaluation of RBC morphology. Representative SEM images showing human RBCs with a normal biconcave shape ((**A**), untreated) or, alternatively, with a flattened shape (leptocytes, arrows) after treatment with 10 µM (**B**) and 100 µM (**C**) mancozeb for 24 h. Magnification 3000×.

**Figure 2 antioxidants-14-01274-f002:**
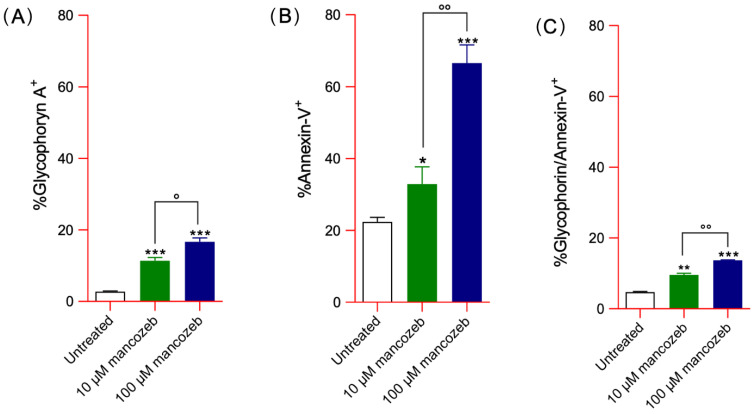
Detection and quantification of extracellular EVs released in RBCs left untreated or treated with 10 µM or 100 µM mancozeb. (**A**), Glycophorin A^+^ EVs; (**B**) Annexin-V^+^ EVs; (**C**) Glycophorin A/Annexin-V^+^ EVs. * *p* < 0.05, ** *p* < 0.01, and *** *p* < 0.001 versus untreated cells; ° *p* < 0.05 and °° *p* < 0.01 between 10 and 100 µM mancozeb, one-way ANOVA followed by Bonferroni’s multiple comparison post hoc test. (*n* = 10). (**D**) Dot plots, obtained from a representative experiment, showing double positivity for glycophorin A and annexin-V in EVs released by RBCs after exposure to both concentrations of mancozeb. The percentage of double-positive vesicles is reported. (**E**–**G**), EV volume analysis by flow cytometry (FACS) based on forward scatter (FSC). The average FSC signal expressed in arbitrary units, reflecting the volume of the EVs in different experimental conditions, is indicated in black within bars. ns, not statistically significant; *** *p* < 0.001 versus EV volume from untreated cells, one-way ANOVA followed by Bonferroni’s multiple comparison post hoc test. (*n* = 10).

**Figure 3 antioxidants-14-01274-f003:**
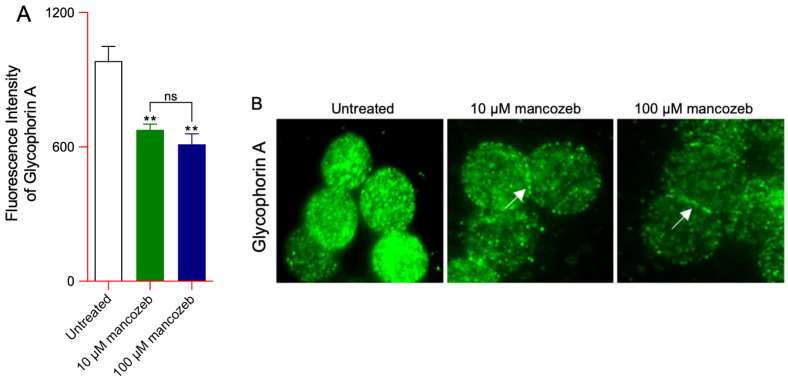
Detection of glycophorin A content and distribution. Human RBCs were left untreated or treated with 10 µM or 100 µM mancozeb for 24 h at 37 °C. (**A**) Histogram reporting mean values of fluorescence intensity of glycophorin A. (**B**) Representative images of immunofluorescence showing glycophorin A distribution in the presence or absence of mancozeb. Arrows indicate a redistributed and clustered along the plasma membrane of glycophorin A. Samples were observed with a 100× objective. ns, not statistically significant; ** *p* < 0.01 versus untreated cells. One-way ANOVA followed by Bonferroni’s post-test; (*n* = 10).

**Figure 4 antioxidants-14-01274-f004:**
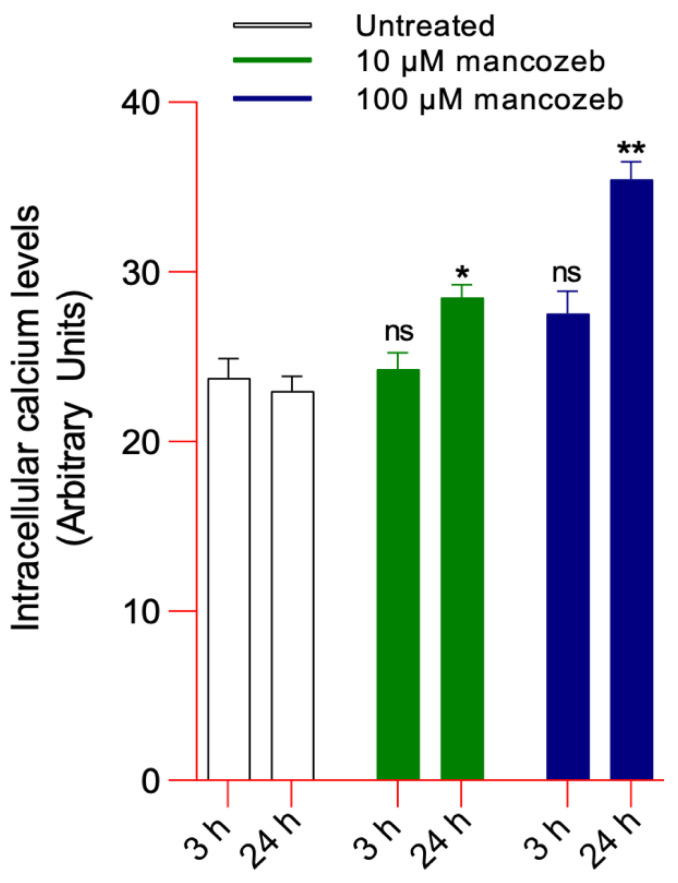
Quantification of intracellular calcium content. Human RBCs were left untreated or treated with 10 µM or 100 µM mancozeb for 24 h at 37 °C. Histograms reporting mean values of fluorescence intensity, representing intracellular calcium levels, as detected by Fluo-3-AM staining. ns, not statistically significant versus untreated cells; * *p* < 0.05, ** *p* < 0.01 versus untreated cells after 24 h incubation. One-way ANOVA followed by Bonferroni’s post-test (*n* = 8).

**Figure 5 antioxidants-14-01274-f005:**
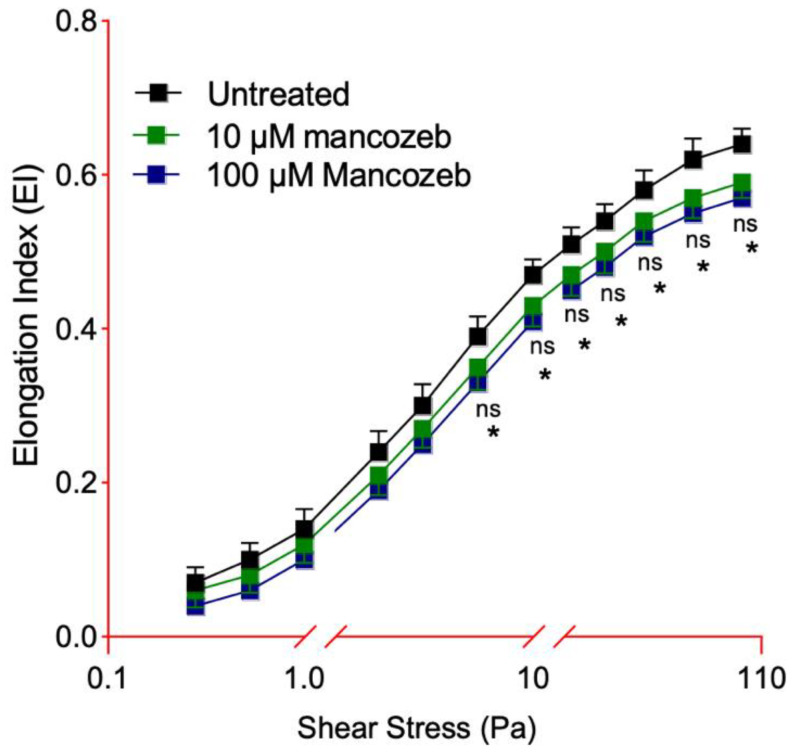
Elongation index of RBCs left untreated or treated with mancozeb at different concentrations (10 and 100 µM), measured across a range of shear stress values. ns, not statistically significant between samples treated with 10 µM and 100 µM mancozeb; * *p* < 0.05, samples treated with 10 µM and 100 µM mancozeb versus untreated RBCs, unpaired Student’s *t*-test (*n* = 8).

**Figure 6 antioxidants-14-01274-f006:**
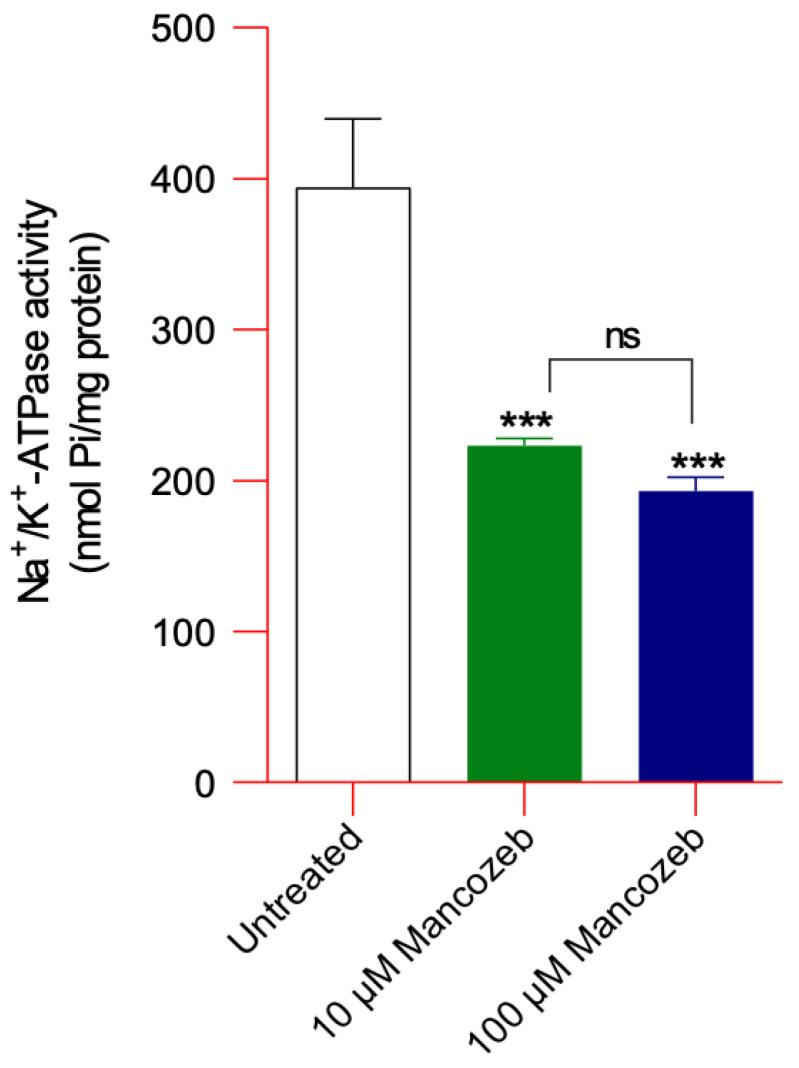
Detection of Na^+^/K^+^ ATPase pump activity in human RBCs exposed to 10 or 100 µM mancozeb for 24 h at 37 °C. ns, not statistically significant; *** *p* < 0.001 versus 10 µM mancozeb. One-way ANOVA followed by Bonferroni’s post-test (*n* = 10).

**Figure 7 antioxidants-14-01274-f007:**
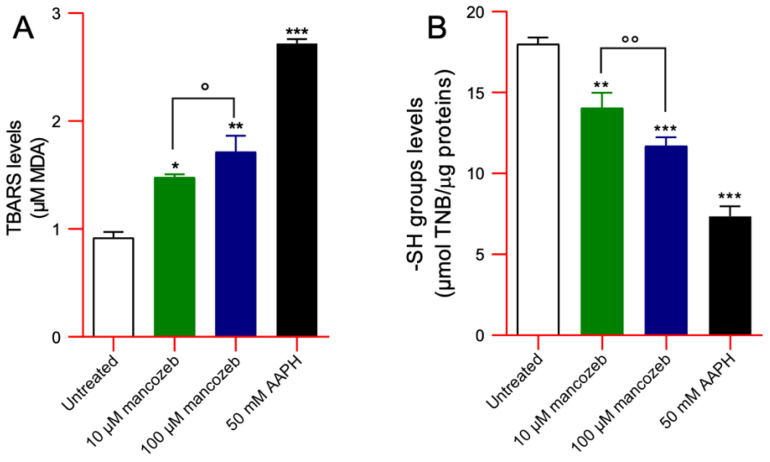
Evaluation of oxidative stress parameters. (**A**) TBARS (µM MDA) and (**B**) content of total sulfhydryl groups (µM TNB/µg protein) were measured in RBCs exposed to 10 or 100 µM mancozeb (24 h at 37 °C), or alternatively, to 50 mM AAPH (1 h at 37 °C); this latter was used as positive control. * *p* < 0.05, ** *p* < 0.01, and *** *p* < 0.001 versus untreated RBCs; ° *p* < 0.05, °° *p* < 0.01 between 10 and 100 µM mancozeb. One-way ANOVA followed by Bonferroni’s post hoc test (*n* = 10).

**Figure 8 antioxidants-14-01274-f008:**
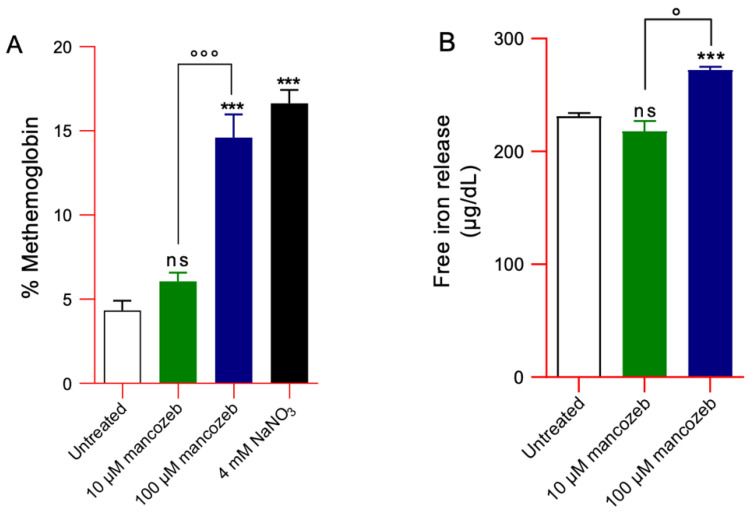
(**A**) Measurement of MetHb levels and (**B**) iron release in cells left untreated or treated with mancozeb. ns, not statistically significant versus untreated cells; *** *p* < 0.001 versus untreated cells; ° *p* < 0.05 and °°° *p* < 0.001 between 10 and 100 µM mancozeb, one-way ANOVA followed by Bonferroni’s post hoc test (*n* = 10).

**Figure 9 antioxidants-14-01274-f009:**
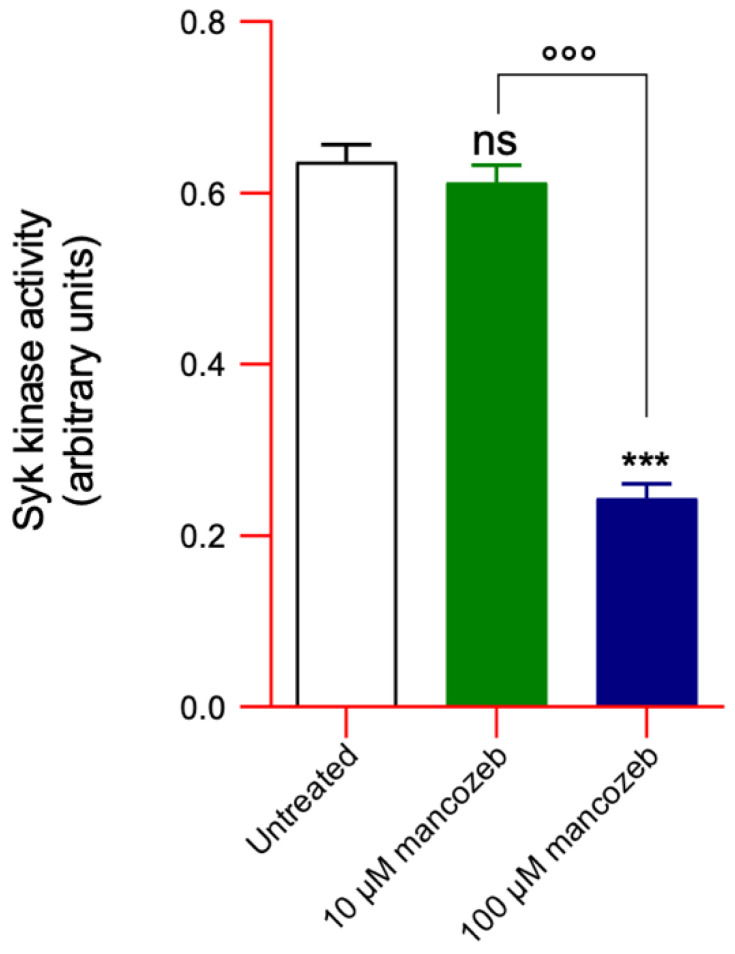
Measurement of cytosolic Syk kinase activity in human RBCs following exposure to 10 µM or 100 µM mancozeb for 24 h at 37 °C. ns, not statistically significant versus untreated cells; *** *p* < 0.001 versus untreated cells; °°° *p* < 0.001 between 10 and 100 µM mancozeb. One-way ANOVA followed by Bonferroni’s post hoc test (*n* = 10).

**Figure 10 antioxidants-14-01274-f010:**
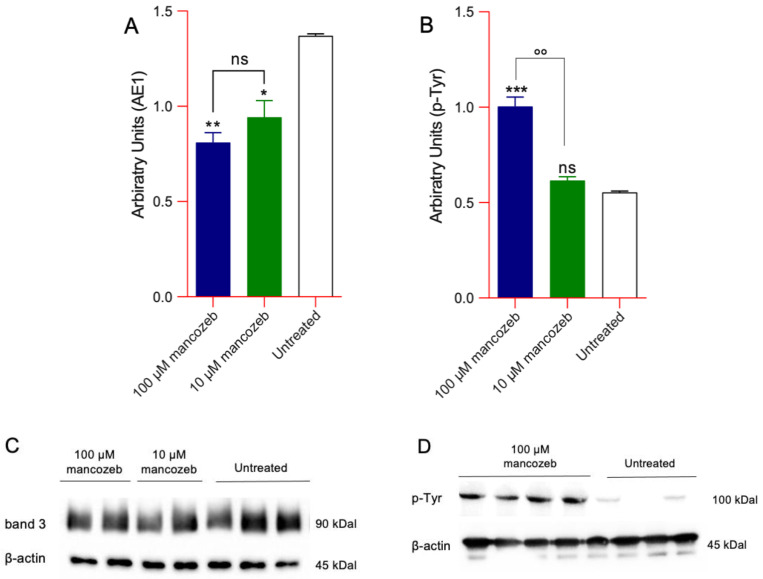
(**A**) AE1 protein abundance and (**B**) p-Tyr levels were detected in the plasma membranes of RBC samples incubated with 10 µM or 100 µM mancozeb for 24 h at 37 °C. (**C**,**D**) Representative Western blotting images are shown. ns, not statistically significant versus untreated RBCs or between 10 and 100 µM mancozeb as indicated, * *p* < 0.05, ** *p* < 0.01, *** *p* < 0.001 versus untreated RBCs, °° *p* < 0.01 between 10 and 100 µM mancozeb. One-way ANOVA followed by Bonferroni’s post hoc test (*n* = 10).

**Figure 11 antioxidants-14-01274-f011:**
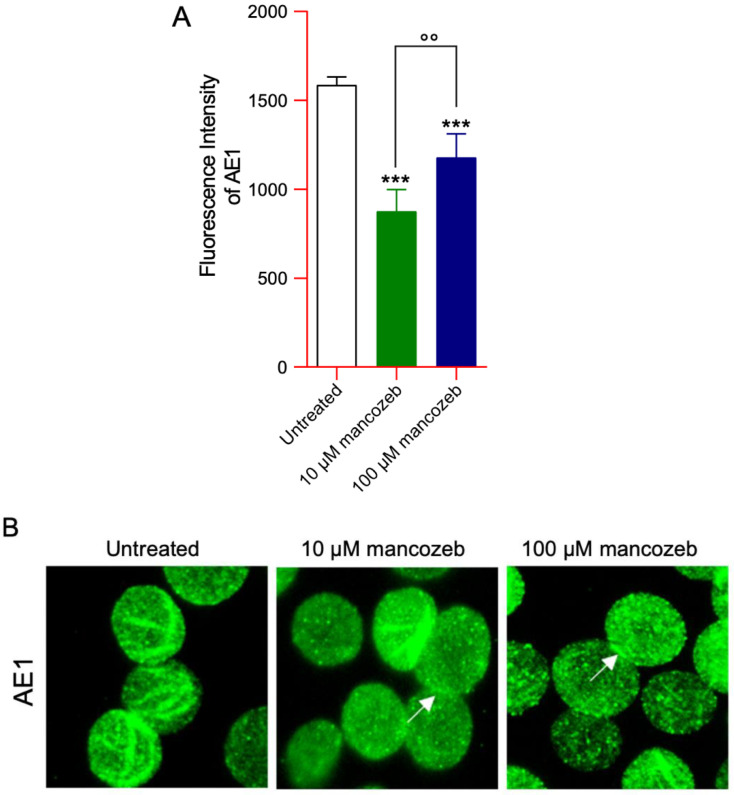
Detection of AE1 protein content and distribution. Human RBCs were left untreated or were treated with 10 or 100 µM mancozeb for 24 h at 37 °C. (**A**) Histograms reporting mean values of fluorescence intensity of AE1. (**B**) Representative images of immunofluorescence showing AE1 distribution in the presence or absence of mancozeb. Samples were observed with a 100× objective. *** *p* < 0.01 versus untreated RBCs; °° *p* < 0.01 between 10 and 100 µM mancozeb; one-way ANOVA followed by Bonferroni’s post-test (*n* = 10).

**Figure 12 antioxidants-14-01274-f012:**
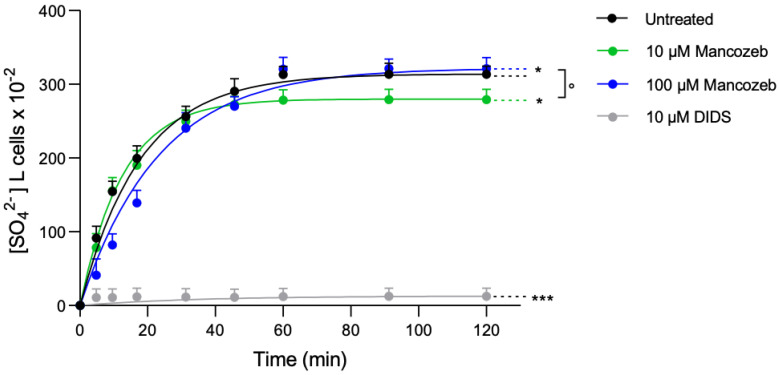
Time course of SO_4_^2−^ uptake in RBCs exposed to 10 µM or 100 µM mancozeb for 24 h at 37 °C or, alternatively, to 10 μM DIDS. * *p* < 0.05 versus untreated cells, *** *p* < 0.001 versus 10 μM DIDS and ° *p* < 0.05 between 10 and 100 µM mancozeb. Two-way ANOVA followed by Bonferroni’s post-test (*n* = 10).

**Figure 13 antioxidants-14-01274-f013:**
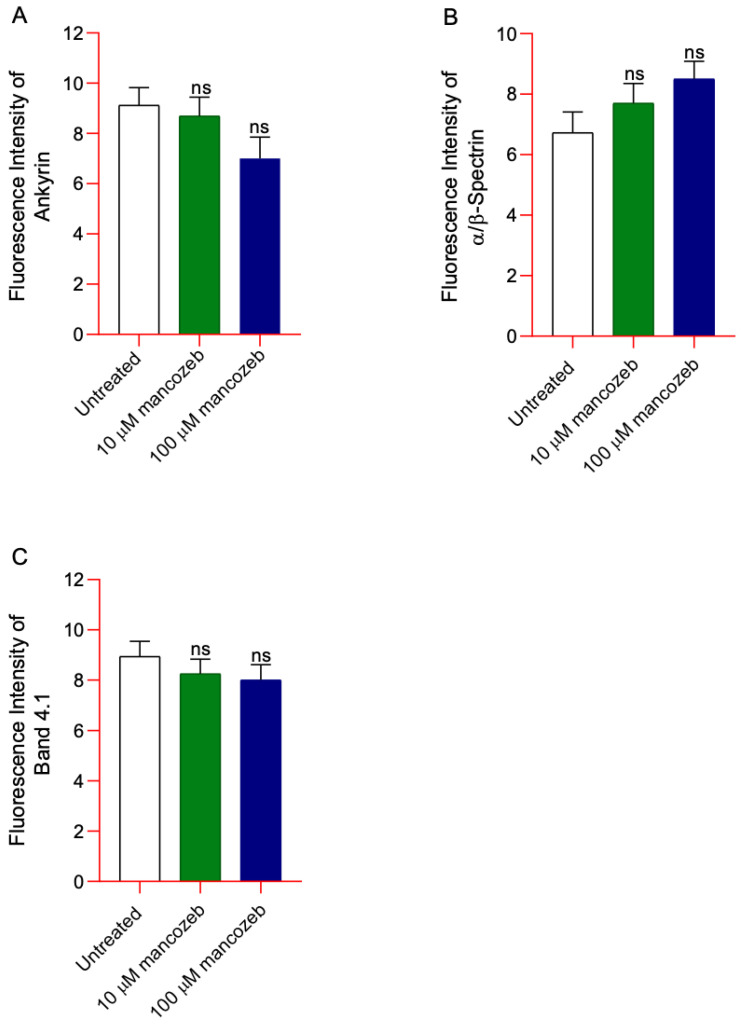
Detection of cytoskeleton-associated proteins. (**A**–**C**) Histograms reporting mean values of protein fluorescence intensity detected in cells treated with 10 µM or 100 µM mancozeb for 24 h at 37 °C. (**D**) Representative images of immunofluorescence showing the distribution of cytoskeleton-associated proteins. Distribution changes (aggregates) are indicated by red arrows. Samples were observed with a 100× objective. ns, not statistically significant versus untreated RBCs, one-way ANOVA followed by Bonferroni’s post-test (*n* = 10).

**Figure 14 antioxidants-14-01274-f014:**
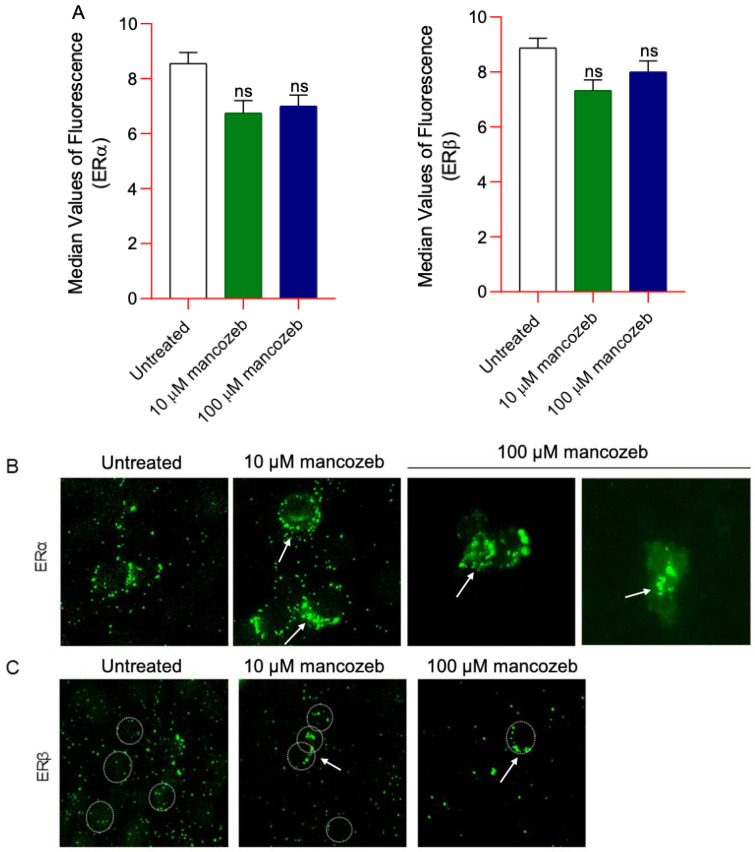
Detection of ERα/β content and distribution. Human RBCs were left untreated or were treated with 10 µM or 100 µM mancozeb for 24 h at 37 °C. (**A**) Histograms reporting mean values of fluorescence intensity of ERs. (**B,C**) Representative immunofluorescence images showing ERα and ERβ distribution in the presence or absence of mancozeb. The dotted white perimeter identifies the RBC cell shape. Samples were observed with a 100× objective. ns, not statistically significant versus untreated RBCs, one-way ANOVA followed by Bonferroni’s post-test (*n* = 8).

**Figure 15 antioxidants-14-01274-f015:**
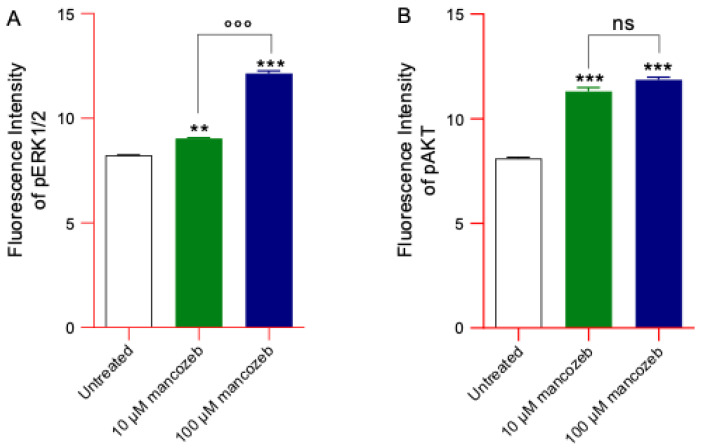
Detection of pERK1/2 and pAKT expression levels. Human RBCs were left untreated or were treated with 10 or 100 mancozeb for 24 h at 37 °C. Histograms report mean values of (**A**) ERK172 an (**B**) pAKT fluorescence intensity. Samples were observed with a 100× objective. ns, not statistically significant versus 10 µM mancozeb; ** *p* < 0.01 and *** *p* < 0.001 versus untreated cells; °°° *p* < 0.001 between 10 and 100 µM mancozeb, one-way ANOVA followed by Bonferroni’s multiple comparison post-test (*n* = 10).

**Figure 16 antioxidants-14-01274-f016:**
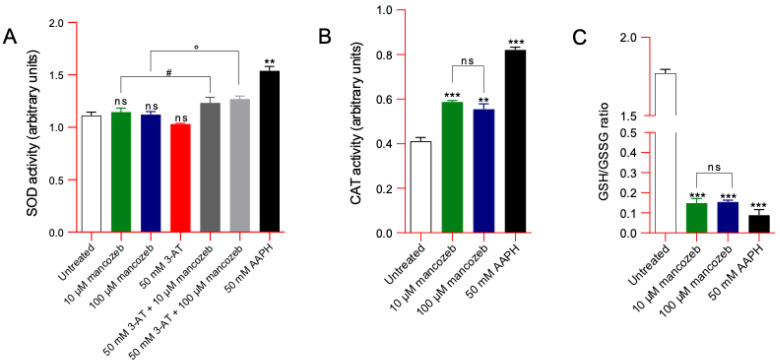
Assessment of the endogenous antioxidant capacity. (**A**) SOD activity, (**B**) CAT activity, and (**C**) GSH/GSSG ratio were detected in RBCs left untreated or treated with 10 or 100 μM mancozeb. Human RBCs were also incubated with 50 mM AAPH (1 h at 37 °C) for the positive control or with 50 mM 3-AT. In a subset of experiments, RBCs were pre-treated with 3-AT and then exposed to mancozeb. ns, not statistically significant versus untreated cells; ** *p* < 0.01 and *** *p* < 0.001 versus untreated cells; # *p* < 0.05 between 10 μM mancozeb and 50 mM 3-AT + 10 μM mancozeb; ° *p* < 0.05 between 100 μM mancozeb and 50 mM 3-AT + 100 μM mancozeb. One-way ANOVA followed by Bonferroni’s post hoc test (*n* = 18).

**Table 1 antioxidants-14-01274-t001:** Percentage of morphological alterations and volume variation (arbitrary units) in cells left untreated or treated as indicated. Data are presented as means ± S.E.M. from three independent experiments, where ns indicates not significant, * *p* < 0.05 and *** *p* < 0.001 versus leptocytes detected in untreated cells. One-way ANOVA followed by Bonferroni’s multiple comparison post hoc test.

Experimental Conditions	Biconcave Shape	Leptocytes	
	%	%	RBC Volume
Untreated	84% ± 0.018	16% ± 0.010	419.5 ± 0.015
10 µM mancozeb	59% ± 0.021	41% ± 0.011 ***	412 ± 0.013 ^ns^
100 µM mancozeb	52% ± 0.022	48% ± 0.09 ***	380.5 ± 0.014 *

**Table 2 antioxidants-14-01274-t002:** Exposure of RBCs to 10 µM or 100 µM mancozeb resulted in a reduction in the volume of EVs. The volume decrease is expressed as a percentage change of the FSC of vesicles from treated cells compared to vesicles from untreated cells (Δ%, arbitrary units). Data are reported as mean ± standard error of the mean (S.E.M.).

	10 µM Mancozeb Δ%	100 µM Mancozeb Δ%
Glycophorin A^+^ EVs	−25 ± 0.021	−23.4 ± 0.017
Annexin-V^+^ EVs	−20 ± 0.025	−18 ± 0.022
Glycophorin A^+^/Annexin-V^+^ EVs	−35.4 ± 0.026	−34.5 ± 0.020

**Table 3 antioxidants-14-01274-t003:** Rate constant of SO_4_^2−^ uptake and SO_4_^2−^ amount internalized in human RBCs exposed to 10 or 100 µM mancozeb for 24 h at 37 °C or, alternatively, to 10 μM DIDS. Results are presented as means ± S.E.M from (*n*) independent experiments; * *p* < 0.05, *** *p* < 0.001 versus untreated, ° *p* < 0.05 versus RBCs treated with 10 µM mancozeb. Two-way ANOVA followed by Bonferroni’s post-test (*n* = 10).

Experimental Condition	Rate Constant (min^−1^)	Time (min)	[SO_4_^2−^] Internalized After 45 min Incubation in SO_4_^2−^ Medium ([SO_4_^2−^] L Cells × 10^−2^)	*n*
Untreated	0.056 ± 0.005	17.71 ±1.76	290.62 ± 5.12	10
10 µM mancozeb	0.079 ± 0.004	12.49 ± 2.02 *	272.5 ± 4.31 ***	10
100 µM mancozeb	0.043 ± 0.005	23.06 ± 1.91 *,°	270 ± 7.15 ***	10
10 µM DIDS	0.030 ± 0.001	35.18 ± 0.89 ***	12.03 ± 0.28 ***	10

## Data Availability

The data that support the findings of this study are available from the corresponding author upon reasonable request.

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
