# Peer review of "Mechanistic Insights into Mancozeb-Induced Redox Imbalance and Structural Remodelling Affecting the Function of Human Red Blood Cells"

_antioxidants, 2025, doi:10.3390/antiox14111274_

Round 1

Reviewer 1 Report

The manuscript by Spinelli et al. describes the effect on RBCs treated by various concentrations of mancozeb in vitro. RBC parameters and changes at the molecular level are assessed by an impressive set of nearly 20 different assay and measurement systems. The selection of methods is rational and the protocols are sound. The presented results give insight in changes at cell level (cell shape, cell volume, deformability, calcium content), at the plasma membrane (Na+/K+-ATPase Activity, AE1 exchange activity, membrane distribution of glycophorin A and AE1 and cytoskeleton proteins), in quantity and composition of shed vesicles, in various cytosolic redox parameters and in signalling pathways (Syk kinase activity, tyrosine phosphorylation of AE1, ERα and ERβ distribution). In the final section, this wealth of findings is reasonably discussed in accordance with widely accepted hypotheses on molecular mechanisms in RBCs and in view of latest developments in RBC research. The authors cite a study by Quds et al from 2023 (DOI: 10.1016/j.pestbp.2023.105453.) which similarly investigates the cytotoxicity on mancozeb-exposed RBCs in vitro. Overlapping assays in both studies are overall corroborative, however, with variations in details, which is per se a valuable information with respect to variability and reproducibility. Apart from that, the study by Spinelli et al. provides additional and complementary information and thereby is a valuable contribution to this research topic.

No further comments

Author Response

We would like to sincerely thank you for the careful evaluation of our manuscript and for the positive comments you have provided. We truly appreciate your recognition of the rationale behind our methodological approach and the relevance of the results within the broader context of red blood cell research. The observations regarding the complementarity of our findings with those of Quds et al., as well as the value of the comparative insights derived from overlapping assays, are particularly appreciated. We are pleased that you consider our contribution meaningful and supportive of current developments in the field.

Reviewer 2 Report

General comments:

The Introduction and Discussion sections should be revised to clearly explain the rationale of the study, the experimental approaches used, and the data they were applied to. For example, the choice of mancozeb concentration range should be justified - currently, the Discussion mentions that a measurable response was obtained within this range, but this information should appear earlier. The same applies to the time intervals: in Section 3.2.2 and Figure 4, a 3-hour interval is presented, but the rationale for this choice is not explained.

Furthermore, much of the content currently placed in the Discussion belongs in the Introduction. Information already presented in the Introduction should not be repeated in the Discussion. Also, the Discussion should not simply restate results without providing additional interpretation or insight.

A separate paragraph discussing the study’s limitations would be very useful. It is also unclear why mancozeb, rather than ethylenethiourea, was chosen as the main compound of interest.

In several figures, the y-axis is labeled only with units instead of the parameter name. I suggest labeling all y-axes consistently, using the parameter name followed by the units in brackets.

Finally, some sections in the Results are not numbered consistently; please review and correct this.

Line 59 – incorrect form of citation

Line 59 – please specify the time period during which the deaths and diseases occurred. Does this happen every year?

Please clarify why the concentration of mancozeb varies between the different methods.

Line 110 – check if correct: Sigma (Milan, Italy)?

Line 130 – were all incubations carried out at 35% HCT?

Line 144 – The control RBC were analyzed by SEM as well.

Line 169 - check the preciseness of formula - “Vtreated − Vuntreated” should be in brackets

Line 189 - The RBC deformability may not reach a plateau at a shear stress of 50 Pa in damaged cells; therefore, the calculated maximal elongation may be inaccurate.

Line 210 - revise: H2DCFDA (subscript)

Line 248 – The manuscript states that Hb oxidation was performed at 25°C; however, in the Results section (line 498), a temperature of 37°C is mentioned. Please verify and correct this discrepancy.

Line 382 - RBC volume was determined by the use of SEM? How? What are units for RBC volume in Table 1?

Line 407 - Legend to the Figure 2 - quantification is presented in % - so, the proportion is measured.

Line 433 - Box plots, not histograms, are presented. Moreover, when the median value is shown, please clarify what the error bars represent — are they SEM? Typically, the interquartile range corresponds to the median. Please also ensure this is corrected consistently in the other figures.

Line 419 - The data presented in Table 2 seem to duplicate those shown in Figure 2E–G. Please consider whether presenting the same data in both formats is necessary.

Line 560 - Figure 12 - is there really significant difference (***) between 100 μM mancozeb and untreated cells?

Line 580 In Figure 13D, the background levels in the representative images (e.g., for spectrin) appear quite different, which could affect the assessment of “distribution changes.”

Line 626 - It is not possible to identify whether an increase of GSSG levels and/or a decrease of GSH levels contribute to a reduction of the GSH/GSSG ratio?

Line 664 – What are the concentrations of mancozeb in the blood of people exposed? Were there any studies focusing on RBC parameters on workers exposed to fungicides, such as Mancozeb?

In abstract, ROS abbreviation is not explained. Each abbreviation should be explained in the text and figure legends (not only in abstract) - e.g. ER abbreviation should also be explained in the text and figure 14. In the figure description, also MVs (fig 2) and AE1 (fig 10) abbreviations should be explained.

Author Response

The Introduction and Discussion sections should be revised to clearly explain the rationale of the study, the experimental approaches used, and the data they were applied to. For example, the choice of mancozeb concentration range should be justified -currently, the Discussion mentions that a measurable response was obtained within this range, but this information should appear earlier. The same applies to the time intervals: in Section 3.2.2 and Figure 4, a 3-hour interval is presented, but the rationale for this choice is not explained. We thank the Reviewer for his/her suggestions. The Introduction and Discussion sections have been slightly revised for clarity. In particular, the rationale for the mancozeb concentrations used—selected to induce measurable cytotoxicity in erythrocytes and allow investigation of oxidative damage mechanisms—has been moved from the Discussion to the Introduction. To assess intracellular calcium levels, a time course was performed at t0h, t3h, and t24h, allowing us to capture both rapid changes in intracellular calcium and longer-term alterations over 24 hours.

Furthermore, much of the content currently placed in the Discussion belongs in the Introduction. Information already presented in the Introduction should not be repeated in the Discussion. Also, the Discussion should not simply restate results without providing additional interpretation or insight. To avoid repetition of information already presented in the Introduction. We thank the Reviewer for this comment. The redundant sections have been removed from the Discussion, and a new paragraph has been added to summarize the main findings, specifically highlighting how the structural and oxidative changes observed in mancozeb-treated RBCs may compromise their function.

A separate paragraph discussing the study’s limitations would be very useful. It is also unclear why mancozeb, rather than ethylenethiourea, was chosen as the main compound of interest. We thank the Reviewer for for raising this point. Mancozeb is the active ingredient applied in agriculture, so studying it allows observation of the effects of the compound in its original form, including potential combined effects of its chemical groups. ETU is only one of the degradation products and therefore does not represent the full chemistry of the pesticide. In humans, exposure initially occurs to mancozeb, which is rapidly metabolized into ETU. Studying mancozeb allows investigation of the cellular damage caused by the actual compound before metabolism, including effects on the membrane, ROS generation, and enzyme activity. ETU is formed only after systemic metabolism; in vitro, cells do not metabolize mancozeb in the same way as a whole organism. Using ETU in vitro may therefore fail to faithfully reproduce the primary toxic effects of mancozeb on the target cells. A sentence has been added in the Introduction section.

Finally, some sections in the Results are not numbered consistently; please review and correct this. Done.

Line 59 – incorrect form of citation. Done.

Line 59 – please specify the time period during which the deaths and diseases occurred. Does this happen every year? We thank the reviewer for this comment. In 1990, a WHO task force estimated that approximately one million unintentional pesticide poisonings occurred each year, resulting in about 20,000 deaths. More than 30 years later, updated global data indicated that every year approximately 355 million cases of unintentional acute pesticide poisoning are expected to occur, leading to around 11.000 deaths annually. Thus, the estimates of deaths and diseases caused by pesticide exposure refer to specific time periods and reflect annual trends (DOI: 10.3390/ijerph18168307). 

Please clarify why the concentration of mancozeb varies between the different methods. We thank the reviewer for this comment. The preliminary phase of the study was designed to test a wide range of mancozeb concentrations (0.5–100 µM) to exclude any overt cytotoxic effects on RBC integrity during a 24 h incubation period. None of the tested concentrations induced detectable hemolysis (Supplementary Materials, Figure S1) and were therefore considered suitable for further investigation. Subsequently, we assessed whether these non-hemolytic concentrations could induce oxidative stress by measuring intracellular reactive oxygen species (ROS) levels in RBCs after 24 h exposure (Supplementary Materials, Figure S3). A significant increase in ROS production was observed within the 10–100 µM range, and based on these results, the lowest and highest concentrations associated with an oxidative response (10 µM and 100 µM, respectively) were selected for the subsequent analyses. Note that varying mancozeb concentrations were used only in these preliminary experiments.

Line 110 – check if correct: Sigma (Milan, Italy)? Thank you for pointing this out. We have corrected the text accordingly, replacing “Sigma (Milan, Italy)” with “Sigma-Aldrich (Milan, Italy)”

Line 130 – were all incubations carried out at 35% HCT? We thank the Reviewer for this question. There was an error in the text: all samples were actually incubated at a hematocrit of 3%, not 35%. The 3% hematocrit was used throughout the experiments.

Line 144 – The control RBC were analyzed by SEM as well. Control RBC were also analyzed by SEM.

Line 169 - check the preciseness of formula - “Vtreated − Vuntreated” should be in brackets. Thank you for pointing this out. We have carefully rechecked the formula and confirm that it is correct as written.

Line 189 - The RBC deformability may not reach a plateau at a shear stress of 50 Pa in damaged cells; therefore, the calculated maximal elongation may be inaccurate. We thank the Reviewer for this comment and partially agree. Although RBC deformability in damaged cells may not fully reach a plateau at a shear stress of 50 Pa, our experimental conditions show a measurable decrease in erythrocyte deformability that still allows for the observation of a partial plateau, rather than a complete absence.

Line 248 – The manuscript states that Hb oxidation was performed at 25°C; however, in the Results section (line 498), a temperature of 37°C is mentioned. Please verify and correct this discrepancy. This parameter has been corrected.

Line 382 - RBC volume was determined by the use of SEM? How? What are units for RBC volume in Table 1? We thank the Reviewer for raising this point. The RBC volume reported in Table 1 was determined by flow cytometry. The values are expressed in arbitrary units, as they represent the mean forward scatter (FSC) signal, which provides a relative measure of cell size rather than an absolute volume.

Line 407 - Legend to the Figure 2 - quantification is presented in % -so, the proportion is measured. Quantification of extracellular vesicles released in RBCs is expressed as %.

Line 433 - Box plots, not histograms, are presented. Moreover, when the median value is shown, please clarify what the error bars represent —are they SEM? Typically, the interquartile range corresponds to the median. Please also ensure this is corrected consistently in the other figures. We thank the Reviewer for raising this point. The values shown are mean values represented as histograms. The legend has been revised accordingly as follows: “Histogram reporting mean values of fluorescence intensity of glycophorin A. All data are expressed as arithmetic mean ± S.E.M.” This correction has also been applied consistently to the other figures.

Line 419 - The data presented in Table 2 seem to duplicate those shown in Figure 2E–G. Please consider whether presenting the same data in both formats is necessary. We thank the Reviewer for raising this point. However, we believe that presenting the data in Table 2 is necessary, as it directly reports the calculated Δ%, which is not explicitly shown in Figure 2E–G.

Line 626 - It is not possible to identify whether an increase of GSSG levels and/or a decrease of GSH levels contribute to a reduction of the GSH/GSSG ratio? We thank the Reviewer for this observation. It is indeed possible to quantify both total glutathione (GSH + GSSG) and GSSG (GSH assay kit, MAK440, Sigma-Aldrich, Milan, Italy). Therefore, the levels of reduced GSH can be indirectly determined by subtracting the measured GSSG from the total glutathione. This approach allows us to infer whether changes in the GSH/GSSG ratio are due to an increase in GSSG, a decrease in GSH, or both.

Line 664 –What are the concentrations of mancozeb in the blood of people exposed? Were there any studies focusing on RBC parameters on workers exposed to fungicides, such as Mancozeb? We thank the Reviewer for this observation. To our knowledge, no studies have reported the concentration of mancozeb in the blood of exposed individuals. This is likely due to its rapid systemic metabolism and degradation into secondary products, such as ethylenethiourea (ETU) [1], which is commonly used as an indirect biomarker of exposure and is measured in urine rather than in blood [2,3]. Indeed, several studies have shown that agricultural workers exposed to mancozeb exhibit increased urinary ETU levels, which correlate with markers of oxidative stress in serum [3,4]. The concentrations of mancozeb used in our study are therefore not directly comparable to human internal exposure levels. However, these concentrations were intentionally selected because they induce measurable cytotoxicity in erythrocytes, allowing us to investigate the mechanisms of oxidative damage. It is also worth noting that the commercial formulations used in agriculture typically contain 1.5–2.0 g/L mancozeb (approximately 2.8–3.7 mM), indicating that occupational settings may also involve contact with relatively high concentrations, even though systemic levels in humans remain low due to rapid metabolism [5].

Colosio, C.; Fustinoni, S.; Birindelli, S.; Bonomi, I.; De Paschale, G.; Mammone, T.; Tiramani, M.; Vercelli, F.; Visentin, S.; Maroni, M. Ethylenethiourea in urine as an indicator of exposure to mancozeb in vineyard workers. Toxicol Lett 2002, 134, 133-140, doi:10.1016/s0378-4274(02)00182-0.

  1. Panganiban, L.; Cortes-Maramba, N.; Dioquino, C.; Suplido, M.L.; Ho, H.; Francisco-Rivera, A.; Manglicmot-Yabes, A. Correlation between blood ethylenethiourea and thyroid gland disorders among banana plantation workers in the Philippines. Environ Health Perspect 2004, 112, 42-45, doi:10.1289/ehp.6499.
  2. Costa, C.; Teodoro, M.; Giambo, F.; Catania, S.; Vivarelli, S.; Fenga, C. Assessment of Mancozeb Exposure, Absorbed Dose, and Oxidative Damage in Greenhouse Farmers. Int J Environ Res Public Health 2022, 19, doi:10.3390/ijerph191710486.
  3. Dall'Agnol, J.C.; Ferri Pezzini, M.; Suarez Uribe, N.; Joveleviths, D. Systemic effects of the pesticide mancozeb - A literature review. Eur Rev Med Pharmacol Sci 2021, 25, 4113-4120, doi:10.26355/eurrev_202106_26054.
  4. Atreya, K.; Sitaula, B.K. Mancozeb: growing risk for agricultural communities? Himalayan Journal of Sciences 2010, 6, 9-10.

In abstract, ROS abbreviation is not explained. Done.

Each abbreviation should be explained in the text and figure legends (not only in abstract) - e.g., ER abbreviation should also be explained in the text and figure 14. In the figure description, also MVs (Fig 2) and AE1 (Fig 10) abbreviations should be explained. We thank the Reviewer for this comment. All abbreviations were explicitly defined at their first appearance in the main text.

Line 560 - Figure 12 - is there really significant difference (***) between 100 μM mancozeb and untreated cells? We thank the Reviewer for this observation. There was a typographical error in the Figure 12; the difference between untreated cells and cells treated with 100 μM mancozeb is p< 0.05 (*), as correctly reported in the table. The graph has been updated accordingly.

Line 580 In Figure 13D, the background levels in the representative images (e.g., for spectrin) appear quite different, which could affect the assessment of “distribution changes.” We thank the Reviewer for this observation. Figure 13 D has been partially modified.

In several figures, the y-axis is labeled only with units instead of the parameter name. I suggest labeling all y-axes consistently, using the parameter name followed by the units in brackets. We thank the Reviewer for this observation. In line with the suggestion, several figures have been revised, and the y-axes have been updated to include both the parameter name and the corresponding units.

Reviewer 3 Report

Detail studies were performed.

See attached comments.

Author Response

The article -Mechanistic Insights into Mancozeb-Induced Redox Imbalance and Structural Remodelling Affecting the Function of Human Red Blood Cells- depicting in vitro studies is very vast and well written. Lot many experiments were performed to prove the fungicide induced redox imbalance in human RBCs. We thank the Reviewer for their positive comments and appreciation of our work. We are glad that the scope of our in vitro studies and the experimental approach to investigate mancozeb-induced redox imbalance in human RBCs were found thorough and well-presented.

1) From quantitative perspective, under -materials and methods- authors need to provide for each experiment the number of RBCs used in terms of % hematocrit value. Also wherever appropriate the amount of RBC total protein content or the amount of hemoglobin used in micro molar amounts. This will give an idea about
the target: fungicide ratio used in the experiment. Also incorporate this information wherever applicable in the ‘figure legends. We thank the Reviewer for this comment. In agreement with Reviewer 2, we have now included the hematocrit percentage used to incubate the cells with mancozeb in the Materials and Methods section.

2) Authors need to provide, say as figure one, schematic of mancozeb conversion to ETU. Such structural visual will also help visualize the binding of mancozeb to specific amino acids in the protein e.g., to the -SH containing ones. We appreciate the Reviewer suggestion to include a schematic illustrating the conversion of mancozeb to ethylenethiourea (ETU). However, we believe that such a diagram is not necessary, as this metabolic pathway is extensively documented in the literature. For instance, a detailed degradation pathway of mancozeb, including the formation of ETU, is available in the publication by Cecconi et al., (2007). In line with Reviewer 2 recommendation, we have incorporated an additional citation in the main text to provide readers with direct access to this information.

3) Mature RBCs in the blood stream are a nucleated. Not sure if RBCs contain RNA and protein expression assembly. Some of the experiments show effect of mancozeb on RBC protein/ enzyme expression levels. Please clarify. We thank the Reviewer for this comment. Mature RBCs are indeed anucleated and, therefore, do not synthesize new RNA or proteins. In this context, when we refer to protein “expression,” we mean the existing protein levels in the cell, i.e., their presence or abundance, rather than de novo synthesis. To name just a few examples: 1) the proteins we analyzed may be partially lost through vesiculation events, resulting in lower levels compared to control conditions; 2) proteins such as AE1 can undergo oxidative stress–induced phosphorylation, which may lead to an apparent increase in protein levels; 3) The observed increase in Syk levels likely reflects its translocation from the cytosol to the membrane, facilitating interaction with membrane-bound AE1, and this redistribution may account for its apparent increase. For the enzymes analyzed in our study, we specifically refer to their activity rather than expression.

4) Under discussion section, line 730, authors hypothesize that hemoglobin oxidation may induce structural changes in AE1, but I think binding of mancozeb directly to AE1 or other protein -SH groups may alter the stability of plasma membrane. We thank the Reviewer for this comment. We agree that, in addition to hemoglobin oxidation, direct binding of mancozeb to AE1 or other -SH containing proteins could potentially contribute to alterations in plasma membrane stability.

5) Other suggestions: Under M&M, a) specify pH of PBS used (Done); b) if measuring absorbance at a specific wave length, say by spectrophotometer (We thank the Reviewer for the comment. The type of instrument used for absorbance measurements has been specified throughout the manuscript, including the spectrophotometer model); c) it’s hemolysis/ hematocrit and not haemolysis/ haematocrit (We thank the Reviewer for the observation. We have intentionally used the British English spelling (haemolysis/haematocrit) throughout the manuscript).

d) line 180, it should be data is presented. Thank you for the clarification. Although “data are presented” is commonly used in scientific writing when treating data as a plural noun, we are pleased to follow the Reviewer preference and have replaced it with “data is presented” as suggested.

  1. e) line 358, it should be as a positive control. Done.